# *O*-GlcNAc and phosphorylation modifications on HtL1/FBA10 regulate wheat vernalization for flowering

Pengfang Yang[1,2,7], Yangyang Liu[1,3,7], Qi Dong[1,2,7], Yuting Miao[1,2], Jianlong Zhang[1,2], Shujuan Xu[1,2,4], Hong Zhao[1,2], Yuda Niu[3,5], Xueyong Zhang [6], Yunyuan Xu [3,5], Zifeng Guo [1,2,3] ✉, Lijing Xing [3,5] ✉ & Kang Chong [1,2,3,5] ✉

Vernalization-regulated flowering is vital for wheat yield and geographical distribution, and the diversity of flowering time genes is essential for the breeding of climate-resilient varieties. Sugars have long been recognized in regulating flowering; however, the intrinsic connection between carbohydrate metabolism and vernalization response remains largely unexplored. Here, we identify a fructose 1,6-bisphosphate aldolase (FBA) encoding gene, *HtL1/FBA10*, as a modulator of heading time variation based on a genome-wide association study utilizing wheat core germplasm collections. Evolutionary analysis shows a decrease in the proportion of haplotype-2 of *HtL1*, which is linked to delayed flowering, in Chinese and American wheat varieties compared to landraces. Vernalization reduces HtL1/FBA10 phosphorylation levels and increases its *O*-GlcNAcylation, which in turn enhances its enzymatic activity and facilitates *VERNALIZATION 1* (*VRN1*) transcription by regulating histone acetylation at the *VRN1* locus. Our findings provide mechanistic insights into the interplay between glucose metabolism and the epigenetic regulation of vernalization in winter wheat.

Wheat (*Triticum aestivum* L.) is the most extensively cultivated staple crop worldwide and constitutes a vital source of caloric intake for the global population[1]. The growing global population and the effects of climate change present significant challenges to future wheat yield[2,3]. Wheat is a heterohexaploid species formed through two natural hybridization events during domestication[4,5]. Flowering timing is vital for grain yield in crops, as it affects environmental conditions during grain filling and impacts reproductive success. Vernalization is crucial for overwintering plants, enabling them to flower after sustained exposure to low temperature[6]. The gene *VRN1* plays a critical role in the genetic network of vernalization response for flowering in wheat, and its expression is contingent upon prolonged cold exposure. *VRN1* encodes a MADS-box transcription factor that acts as an integrator to promote flowering[7]. In temperate cereals like wheat and barley, vernalization-induced expression of *VRN1* is intricately regulated by epigenetic modifications[8–10]. Prior to vernalization, the repressive histone marker H3K27me3 is deposited at the *VRN1* locus to inhibit its transcription in barley and wheat. During vernalization, the active markers of H3K4me3 and H3K36me3 gradually accumulate, while H3K27me3 levels decline at *VRN1*[8,9]. Short-term cold exposure quickly increases histone acetylation at the *HvVRN1* locus, activating its transcription through a mechanism similar to the regulation of cold acclimation genes[11]. However, little is known about the regulatory mechanisms governing histone acetylation at *VRN1* in response to prolonged cold for flowering in cereals.

[1]State Key Laboratory of Forage Breeding-by-Design and Utilization, Institute of Botany, Chinese Academy of Sciences, Beijing, China. [2]University of Chinese Academy of Sciences, Beijing, China. [3]China National Botanical Garden, Beijing, China. [4]Institute of Science and Technology Austria, Klosterneuburg, Austria. [5]Key Laboratory of Plant Molecular Physiology, Institute of Botany, Chinese Academy of Sciences, Beijing, China. [6]Institute of Crop Sciences, Chinese Academy of Agricultural Sciences, Beijing, China. [7]These authors contributed equally: Pengfang Yang, Yangyang Liu, Qi Dong.
✉e-mail: guozifeng@ibcas.ac.cn; xinglijing@ibcas.ac.cn; chongk@ibcas.ac.cn

*O*-linked β-D-*N*-acetylglucosamine (*O*-GlcNAc) modification of proteins, reversibly catalyzed by *O*-GlcNAc transferase (OGT) and *O*-GlcNAcase (OGA)[12], has extensive interplay with phosphorylation[13]. *O*-GlcNAcylation and phosphorylation modulate the physiological functions of proteins through synergistic or antagonistic interactions, with the crosstalk between these two post-translational modifications exerting a critical regulatory influence on the pathogenesis of human diseases[14–16]. *O*-GlcNAc transferases play a role in the regulation of flowering in plants[17,18]. In winter wheat, proteins undergo dynamic post-translational phosphorylation and *O*-GlcNAcylation modifications during the vernalization process[19]. The vernalization gene *VRN-D4*, predominantly identified in ancient wheat subspecies from South Asia, represents a duplicated copy of *VRN-A1*. The RIP-3 region within the first intron of *VRN-D4* shows similarity to the binding site of the *Arabidopsis* GLYCINE-RICH RNA-BINDING PROTEIN7 (GRP7), which is known to regulate flowering[20,21]. Three close SNPs in RIP-3 region of *VRN-D4* and certain *VRN-A1* alleles are linked to vernalization requirements, and variations of these three SNPs influence the binding capacity of GRP2, a wheat homolog of GRP7, to the RIP-3 region[20]. The binding of GRP2 to RIP-3 is also affected by its post-translational *O*-GlcNAcylation[19], which is catalyzed by TaOGT1[18]. Therefore, *O*-GlcNAc modification of proteins plays a potential role in vernalization gene-mediated flowering time regulation. Phosphorylation of histone H3 and RNA polymerase II at the *FLC* locus modulates *FLC* transcription, affecting flowering transition[22,23]. Nevertheless, there are a few examples of a single protein exhibiting both *O*-GlcNAcylation and phosphorylation, referred to as "yin-yang" modifications[24], that regulate plant development.

A strong correlation exists between sugar metabolism and epigenetic inheritance in cancer cells. Previous research has demonstrated that metabolites generated through the glycolytic pathway function as substrates or cofactors, playing a significant role in the regulation of epigenetic modification processes[25]. For instance, acetyl-coenzyme A (acetyl-CoA), serving as the substrate for histone acetylation modification enzymes, influences the levels of histone acetylation mediated by these enzymes within cells[26]. The dynamic fluctuations in glycolytic activity can impact acetyl-CoA levels, subsequently modulating histone acetylation in both yeast and mammals[27,28]. Fructose-1,6-bisphosphate aldolase (FBA) facilitates the cleavage of fructose 1,6-bisphosphate (FBP) into dihydroxyacetone phosphate (DHAP) and glyceraldehyde-3-phosphate (G3P), while also catalyzing the reverse reaction to synthesize FBP. FBA plays a crucial role in the glycolytic and gluconeogenic pathways, thereby regulating energy metabolism and carbon flux[29]. FBA is implicated in various vital biological processes and responds to diverse biotic and abiotic stresses[30,31]. However, there remains limited understanding of how the glycolysis process integrates FBA activity with specific environmental and developmental signaling in plants.

Here, a genome-wide association study (GWAS) was performed to identify key genetic loci and major genes associated with heading time traits in 91 cultivars of the wheat core germplasm collections. The QTL, *quantitative Heading-time Locus1* (*qHtL1*), was identified within 639–641 Mb on chromosome 3 A, in which a predicted FBA encoding gene, *HtL1/FBA10*, was identified as a modulator of flowering time. Vernalization leads to reduced phosphorylation and elevated *O*-GlcNAc modifications of HtL1/FBA10, thereby enhancing its activity. This change subsequently alters histone acetylation levels at the *VRN1* locus, promoting flowering in winter wheat.

## Results

### *TraesCS3A02G391100* was identified as the candidate gene of flowering time

Flowering time is a critical factor influencing wheat grain yield and adaptability[3]. To investigate the genetic variation of heading time

among diverse wheat accessions, phenotypic data of heading time of 91 wheat accessions from different regions over three consecutive years (2014, 2015, and 2016), as well as BLUP value heading time over three years, were combined with resequencing data for GWAS analysis[32]. These accessions comprised 65 modern cultivars, 17 local varieties, and nine elite varieties introduced into China (Supplementary Data 1). The analysis revealed significant association signals across 21 chromosomes, with identified genomic regions containing known flowering-related genes, including *FLOWERING LOCUS T 3* (*FT3*), *FRUITFULL-like 3* (*FUL3*), *GIGANTEA* (*GI*), and *VRN1* (Supplementary Fig. 1). The major locus *qHtL1* associated with flowering was located on chromosome 3A (639–641 Mb) (Fig. 1a, b and Supplementary Fig. 2). A total of 22 genes were located within this region (Supplementary Data 2 and Supplementary Data 3). By analyzing previously published transcriptomic data and publicly accessible databases (http://202.194.139.32/expression/index.html), it was observed that *TraesCS3A02G391100*, which encodes FBA10, displayed a relatively high expression level during the heading stage (Z-65)[4] (Fig. 1c and Supplementary Data 2). Furthermore, *TraesCS3A02G391100/FBA10* exhibited the highest expression level in spikes, and *TraesCS3A02G391200* and *TraesCS3A02G391500* also showed relatively high expression levels in spikes among the 22 genes[4] (Supplementary Fig. 3a and Supplementary Data 2). Further analysis of transcriptome data revealed that *TraesCS3A02G391100/FBA10* consistently maintained higher expression levels across various developmental stages in spikes[33] (Fig. 1d, only genes with FPKM > 10 were shown). *TraesCS3A02G391200* encodes a protein similar to FLC EXPRESSOR (FLX), which is essential for the establishment of vernalization requirement in *Arabidopsis*[34,35], suggesting its potential role in flowering regulation in wheat. Further haplotype analysis of *TraesCS3A02G391200* identified two distinct haplotypes in wheat collections. Phenotypic evaluation revealed no significant differences in heading time among accessions carrying these two haplotypes, indicating the absence of genotype-phenotype association at this locus (Supplementary Fig. 3b). In addition, a haplotype association analysis of *TraesCS3A02G391500*, which is predicted to encode FBA11, revealed no significant differences in heading time between two haplotypes (Supplementary Fig. 3c). Further analysis revealed two haplotypes for *TraesCS3A02G391100/FBA10*, with haplotype 2 (Hap-2) being consistent with the Chinese Spring reference genome. Significant differences in heading time were observed between cultivars with these two haplotypes, and cultivars carrying Hap-1 exhibited an earlier heading time (Fig. 1e and Supplementary Fig. 4). Furthermore, the results of real-time fluorescence quantitative PCR (RT-qPCR) showed that the transcript levels of *TraesCS3A02G391100/FBA10* in Hap-1 group were generally higher than that in Hap-2 (Supplementary Fig. 5). The SNP (S3A_639082983) in the promoter region of *FBA10/HtL1* lies within the predicted binding site for the BBR-BPC transcription factor (Supplementary Data 4), which has been previously validated to regulate gene expression[36]. Although the amino acid sequences of Hap-1 and Hap-2 remain unchanged, this promoter variation may disrupt BBR-BPC binding, potentially altering *FBA10/HtL1* expression. These findings suggested that *TraesCS3A02G391100/FBA10/HtL1* (hereafter referred to as *HtL1*) is the candidate gene potentially associated with this genomic region.

To investigate the geographic distribution and breeding selection of *HtL1*, the haplotypes were analyzed in 306 worldwide wheat accessions[37,38], 355 cultivated varieties bred in different periods[39], and 830 Chinese wheat varieties from different breeding periods[40]. Five main haplotypes (Hap-W1-5) were identified in the 306 worldwide wheat accessions, with Hap-W5 being the predominant haplotype. Hap-W1 predominated in the Middle East and Africa, while Hap-W3 was prevalent in North America (Fig. 1f). Among the 355 landraces and

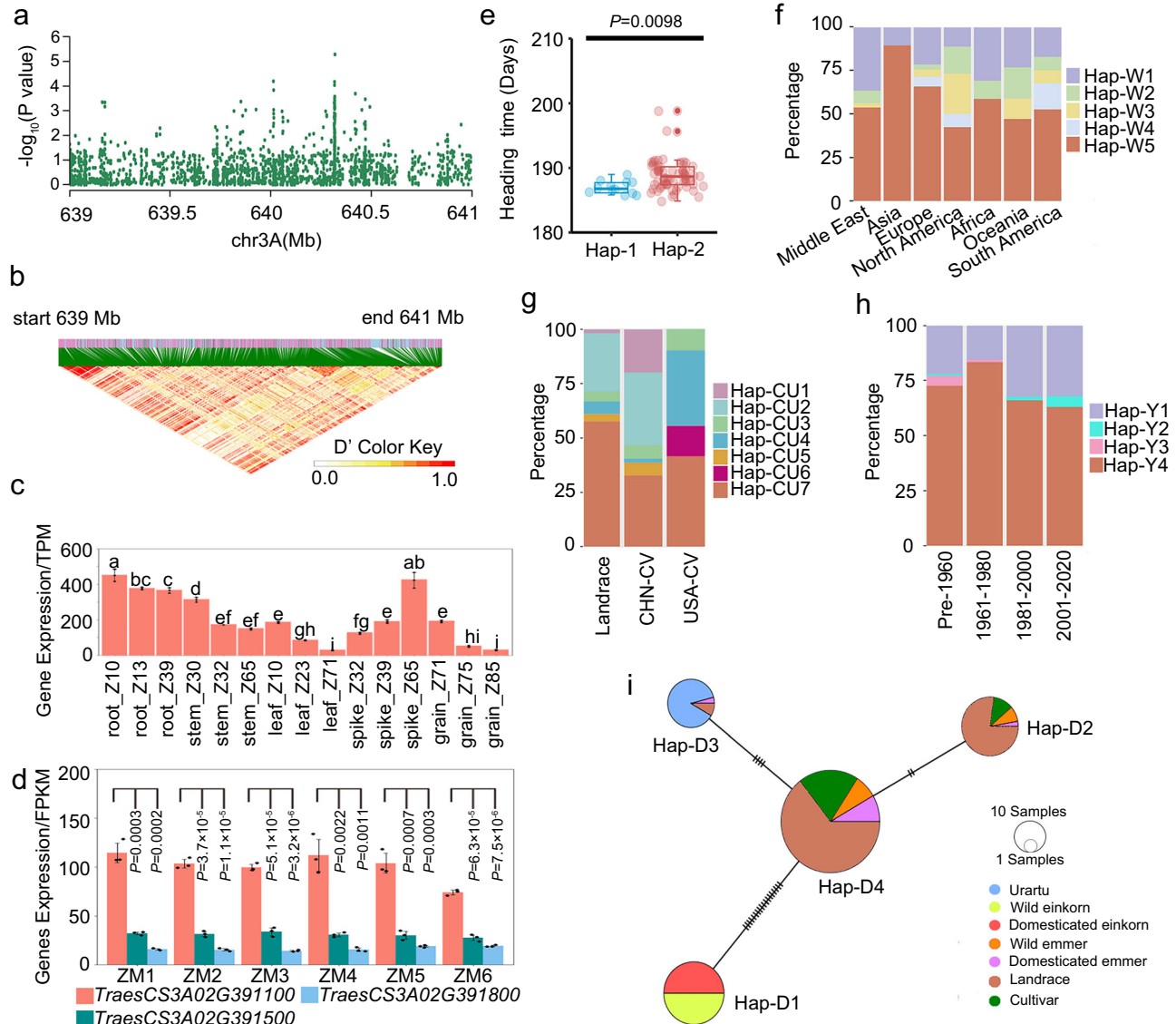

**Fig. 1 | *TraesCS3A02G391100* is associated with flowering traits. a** Manhattan plot of GWAS for response to heading time in the region between 639 and 641 Mb on chromosome 3A. **b** Linkage disequilibrium plot for SNPs. **c** Tissue-specific expression profile of *TraesCS3A02G391100*. Z: Zadoks decimal code, internationally used to represent developmental stages of cereal crops, with different numbers representing distinct developmental stages. The expression levels of the gene in different tissues were downloaded from wheatOmics1.0. Data are presented as mean ± SD (*n* = 2 independent biological replicates, and the third one is the mean of the first two replicates), significance was measured with one-way ANOVA with Tukey test and different letters (**a**–**i**) indicate statistically significant (*P* < 0.05) differences between groups. **d** The expression patterns of *TraesCS3A02G391100* and indicated candidate genes during spikelet development. ZM1-ZM6 represent the stages of one-leaf stage, two-leaf stage, tillering initiation, primordium differentiation of floret, primordium differentiation of spikelet, and maturity, respectively. Data are presented as mean ± SD (*n* = 3 independent biological replicates), two-tailed

Student's *t*-test for statistical analysis. **e** The differences in heading time among materials containing two different Hap-1 (*n* = 12 accessions) and Hap-2 (*n* = 66 accessions) of *TraesCS3A02G391100*. For each box, the upper and lower boundaries represent the 75th and 25th percentile, respectively. The middle horizontal lines represent the median. The whiskers represent 1.5× the interquartile range. The individual data points are plotted as open circles. The solid dots beyond the whiskers represent outliers. Two-tailed Student's *t*-test for statistical analysis. **f** Geographic distribution of the haplotypes of *TraesCS3A02G391100* across 306 worldwide wheat accessions. **g** Haplotype frequency of *TraesCS3A02G391100* among 355 wheat accessions in China and USA cultivars and landraces. CHN-CV: China cultivars, USA-CV: USA cultivars. **h** The haplotype frequency of *TraesCS3A02G391100* among 830 wheat accessions from 1900 to 2020 in China. **i** Haplotype network of *TraesCS3A02G391100*. The SNPs involved in haplotypes Hap-2, Hap-W5, Hap-CU7, Hap-Y4, and Hap-D4 are consistent with the reference genome sequence (Chinese Spring). Source data are provided with this figure.

modern cultivars from China and the United States, seven main haplotypes (Hap-CU1-7) were found. Hap-CU2 and Hap-CU7 were the predominant haplotypes. Hap-CU4 and Hap-CU6 were also mainly distributed among American cultivated varieties. Meanwhile, in Chinese cultivated varieties of the 355 accessions, Hap-CU1 and Hap-CU2 showed higher proportions (Fig. 1g). It indicated a priority selection to these haplotypes in China and the United States, possibly related to its genetic adaptability. We identified four haplotypes (Hap-Y1-4) among

830 cultivars released at different breeding stages. Hap-Y4 was the main haplotype, and its proportion gradually decreased from 1961 to 2020 (Fig. 1h).

Using 206 accessions of different ploidy types (comprising 61 diploid accessions, 22 tetraploid accessions and 123 hexaploid accessions), we investigated the variation of *HtL1* during wheat domestication. Four main haplotypes (Hap-D1-4) were identified, Hap-D1 was predominantly distributed in wild and domesticated einkorn wheat,

while Hap-D2 and Hap-D4 were mainly found in landraces and cultivated varieties, and Hap-D3 was primarily observed in *Triticum Urartu* (Fig. 1i and Supplementary Fig. 6). The results indicated that different haplotypes may have distinct origin backgrounds. Specifically, Hap-D4 is exclusive to tetraploid and hexaploid wheat, indicating a potential formation during wheat polyploidization. It is hypothesized that Hap-D4 could have originated from Hap-D3 in the process of polyploidization.

Previous studies showed that O-GlcNAc modification of proteins regulates sugar metabolism-associated flowering in wheat[18,41]. Protein modification-omics showed that FBA10/HtL1 undergoes O-GlcNAc and phosphorylation modifications in response to vernalization[19]. Therefore, *HtL1* was selected for further investigation to clarify its role in regulating flowering in wheat.

### *HtL1* promotes heading in winter wheat

To investigate the biological function of *HtL1* in flowering regulation, we constructed overexpression lines of *HtL1* by using the constitutive ubiquitin promoter in the winter cultivar KN199 background, thereby enabling the continuous constitutive expression of HtL1-HA in KN199. Meanwhile, we employed the CRISPR-Cas9 gene editing system to knock out all A-, B- and D-homeologs of *HtL1* in KN199 background. The target sites for the single guide RNA (sgRNA) were designed within exon 2. In the T0 generation of transgenic plants, only two *HtL1* knockout mutants (*cr-1* and *cr-2*) were identified, both of which demonstrated characteristics indicative of fragment deletion (Supplementary Fig. 7). Unfortunately, these two mutants exhibited a lethal phenotype under this condition. A relevant study demonstrates that the knockdown of the gene encoding aldolase A results in lethal outcomes in murine cell lines[42]. We further generated *HtL1* knockdown mutants using RNA interference (RNAi) approach, and a specific 273 bp CDS sequence of *HtL1* was selected for precise targeting. Following vernalization, phenotypic analysis of flowering time indicated that the *HtL1* overexpression (*HtL1*-OE) lines exhibited a reduced heading time compared to the KN199 control. Conversely, the *HtL1* knockdown mutants displayed a significantly delayed heading time relative to KN199 (Fig. 2a, b and Supplementary Fig. 8). Based on Unique Identifier mRNA Sequencing (UID mRNA-seq) data derived from wheat plumules of KN199 and *HtL1*-RNAi plants without vernalization treatment, as well as further RT-qPCR analysis of indicated genes at stem elongation stage in *HtL1*-RNAi and KN199 plants with or without vernalization treatment, we evaluated the impact of *HtL1* knockdown on the transcription of other homologous genes, indicating that *HtL1* knockdown achieved the anticipated specificity (supplementary Fig. 9). RT-qPCR results revealed that *HtL1* expression was upregulated in the *HtL1*-OE lines but downregulated in the *HtL1*-RNAi lines compared to KN199 (Fig. 2c). Furthermore, the transcription levels of *VRN1* were elevated in the *HtL1*-OE plants, while decreased in the *HtL1*-RNAi plants relative to KN199 (Fig. 2d). These findings implied that *HtL1* may regulate flowering time in wheat by modulating the expression of *VRN1*.

Subcellular localization analysis showed that HtL1/FBA10 is located in cytosol (Supplementary Fig. 10). Structural prediction performed using SWISS-MODEL indicated that HtL1/FBA10 forms a tetrameric protein structure, comprising four monomeric units (Fig. 2e, f), which aligns with the structural characteristics of class I aldolases[43]. To assess the enzyme activity, the HtL1-GST fusion protein was expressed and subsequently purified from *Escherichia coli* strain BL21 (Supplementary Fig. 11). Enzyme kinetic curves and activity assays collectively demonstrated that HtL1 exhibits fructose-bisphosphate aldolase activity (Fig. 2g, h). Analysis of endogenous aldolase activity in plants demonstrated a significant increase in enzyme activity in *HtL1* overexpression lines relative to the KN199, whereas a significant decrease was observed in *HtL1* knockdown plants (Fig. 2i). Taken together, these results indicate that the

enhancement of aldolase activity of HtL1 facilitates the promotion of flowering in wheat.

### TaCDPK13 phosphorylates HtL1 to inhibit its aldolase activity

To explore the regulatory mechanisms governing HtL1 activity, immunoprecipitation-mass spectrometry (IP-MS) was conducted using an anti-HA antibody to identify proteins that potentially interact with HtL1 in cell lysates derived from *HtL1*-OE plants. The post-translational phosphorylation modification of FBA10/HtL1 was previously observed in response to vernalization[19], indicating that the calcium-dependent protein kinase TaCDPK13, among other candidate interaction proteins, may function as a potential interacting partner of HtL1 (Supplementary Data 5). The interaction between HtL1 and TaCDPK13 was validated by a Luciferase Complementation Imaging (LCI) assay in plant. The results showed obvious luminescence signals when *Nicotiana benthamiana* leaves were co-transformed with *HtL1-cLuc* and *TaCDPK13-nLuc* constructs (Fig. 3a). In addition, the interaction between HtL1 and TaCDPK13 is enhanced in the presence of $Ca^{2+}$ (Fig. 3a). Bimolecular fluorescence complementation (BiFC) analysis revealed robust YFP fluorescence signals in *N. benthamiana* leaves co-expressing HtL1-cYFP and TaCDPK13-nYFP (Supplementary Fig. 12). Co-immunoprecipitation (Co-IP) further conformed this interaction (Fig. 3b). Collectively, these findings demonstrated that HtL1 physically interacts with TaCDPK13 in vivo and in vitro.

To ascertain whether HtL1 functions as a substrate for TaCDPK13, in vitro experiments were performed. Both the phosphorylation of HtL1 by TaCDPK13 and the autophosphorylation activity of TaCDPK13 were confirmed. The phosphorylation modification levels of HtL1 progressively intensify with the increased incorporation of TaCDPK13 (Fig. 3c), which was further confirmed by mass spectrometry (Supplementary Fig. 13a), and the phosphorylation sites on HtL1 catalyzed in vitro by TaCDPK13 were identified (Supplementary Fig. 13b, c). Furthermore, in vivo constitutively expressed HtL1-HA was immunoprecipitated from *HtL1*-OE plants using anti-HA antibody for mass spectrometry analysis, and two overlapped phosphorylation sites were identified (Supplementary Fig. 14). These data confirmed that TaCDPK13 is responsible for HtL1 phosphorylation.

To further determine the alterations in phosphorylation levels of HtL1 during vernalization in winter wheat, the phosphorylation status of HtL1-HA from wheat plumules was immunoblotted using anti-phosphoSer/Thr antibody. When maintaining consistent total protein levels across varying durations of vernalization (V0, V7, V14 and V21), the constitutively expressed HtL1-HA protein exhibited accumulation with prolonged cold exposure, whereas its phosphorylation levels showed a significant reduction (Fig. 3d). To assess the impact of TaCDPK13-mediated phosphorylation on the activity of HtL1, HA-tagged HtL1 was expressed in *N. benthamiana* leaves in the presence or absence of TaCDPK13. The results showed that co-expression of HtL1-HA and GFP-TaCDPK13 led to a reduction in the enzymatic activity of HtL1 when compared to co-expression of HtL1-HA and GFP (Fig. 3e), indicating that the phosphorylation of HtL1 catalyzed by TaCDPK13 inhibits its aldolase activity. The effect of vernalization on in vivo aldolase activity was further evaluated in *HtL1*-OE wheat plants, revealing that vernalization significantly enhanced aldolase activity (Fig. 3f).

Subsequently, we generated overexpression lines of *TaCDPK13* driven by the ubiquitin promoter in KN199 background to investigate the influence of *TaCDPK13* on flowering time. Additionally, we identified two mutants, *tacdpk13-1* and *tacdpk13-2*, from the KN9204 EMS mutant library, which induce premature termination of TaCDPK13 protein translation due to base mutations (Supplementary Fig. 15). Under vernalization conditions, *TaCDPK13* overexpression lines exhibited a delayed flowering phenotype compared to KN199. In contrast, the two mutants showed an earlier flowering phenotype (Fig. 4a, b). RT-qPCR analysis showed that *TaCDPK13* expression was

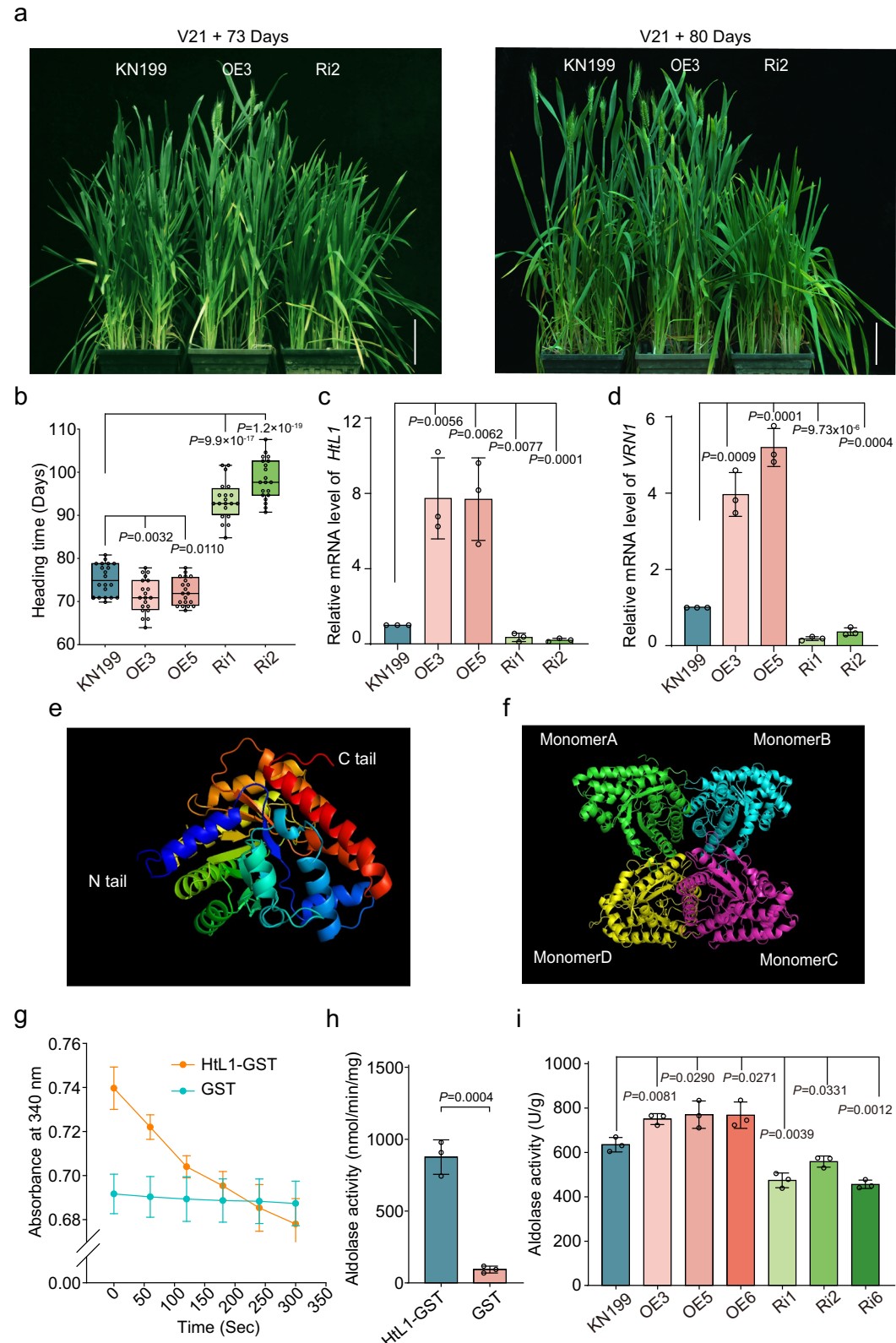

upregulated in the *TaCDPK13*-OE lines relative to KN199 (Fig. 4c). Furthermore, the expression levels of *VRN1* were assessed in the transgenic lines. The transcript abundance of *VRN1* was reduced in *TaCDPK13*-OE plants compared to KN199, whereas it was elevated in *tacdpk13* mutants (Fig. 4d, e). These results indicate that *TaCDPK13* negatively regulates flowering by inhibiting *VRN1* transcription in wheat. To examine the effect of TaCDPK13 on the activity of fructose-

1,6-bisphosphate aldolase in wheat, we quantified the overall aldolase activity in *TaCDPK13*-OE transgenic lines. The results indicated a reduction in activity relative to the KN199 control (Fig. 4f). Incubating cell lysates of *TaCDPK13*-OE plants with in vitro purified HtL1-GST led to significantly higher HtL1 phosphorylation levels compared to KN199 (Fig. 4g). Together these results suggest that TaCDPK13 negatively regulates flowering by phosphorylating HtL1 to inhibit its activity.

**Fig. 2 | *HtL1* accelerates flowering in winter wheat. a** The morphological phenotypes of *HtL1*-OE, *HtL1*-RNAi and wild-type KN199 plants. Seeds were initially germinated and cultivated in the dark at room temperature for 2 to 3 days. Subsequently, the materials were vernalized at 4 °C for 21 days, after which they were transplanted into the greenhouse. Flowering phenotypes of *HtL1*-OE and *HtL1*-RNAi lines were observed and photographed at 73 and 80 days, respectively. Scale bars, 10 cm. **b** Statistical analysis of heading time of *HtL1*-OE, *HtL1*-RNAi and KN199 plants. The box plots display the interquartile range, comprising the first quartile, median, and third quartile, while the whiskers extend from the minimum to the maximum values. Two-tailed Student's *t*-test for statistical analysis, (*n* = 20 plants for each line). **c** and **d** The relative mRNA abundance of *HtL1* and *VRN1* in *HtL1*-OE and *HtL1*-RNAi lines, as well as KN199. RNA were extracted from plumules of wheat

plants vernalized for 21 days, and transcription levels of the indicated genes were analyzed by RT-qPCR. The data were normalized to *ACTIN*, then normalized to KN199. Independent biological experiments were conducted three times. Data are mean ± SD, two-tailed Student's *t*-test for statistical analysis. **e** and **f** The monomeric and homotetrameric structures of HtL1 predicted by SWISS-MODEL. **g** and **h** Enzyme kinetic curve and activity assays of in vitro purified HtL1. Three independent biological replicates were performed. Data are presented as mean ± SD, two-tailed Student's *t*-test for statistical analysis. **i** Aldolase activity analysis in *HtL1*-OE and *HtL1*-RNAi plants with 21 days of vernalization treatment. Three independent biological replicates were performed. Data are mean ± SD, two-tailed Student's *t*-test for statistical analysis.

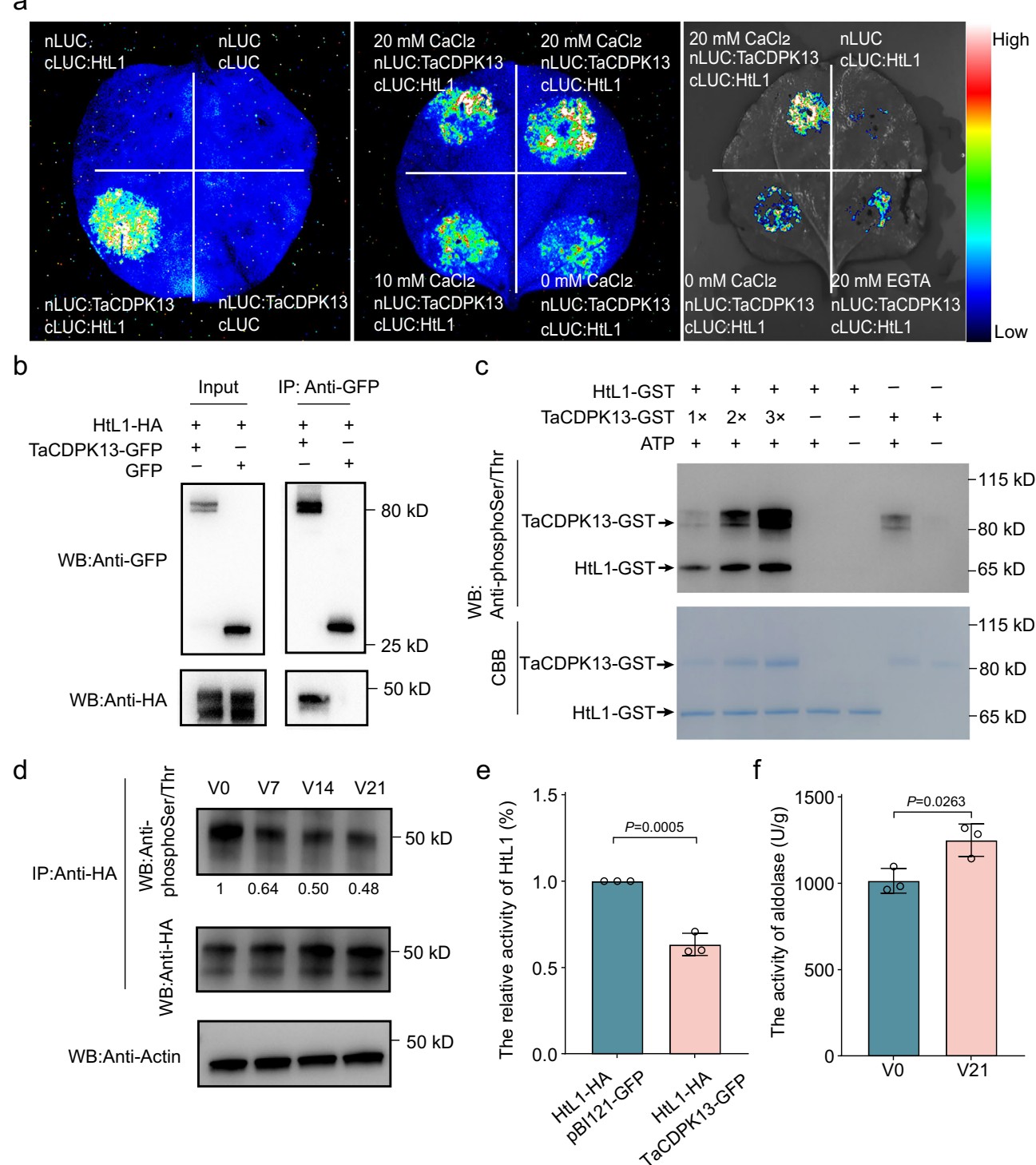

**Fig. 3 | Phosphorylation of HtL1 by TaCDPK13 inhibits its enzyme activity.**
**a** Detection of the interaction between HtL1 and TaCDPK13 using Luminescence Complementation Imaging (LCI) assay. CaCl₂ enhances the interaction between HtL1 and TaCDPK13, whereas EGTA reduces it. Luciferase activity was assessed 48 h after infiltration. Three independent biological replicates were performed. **b** Co-immunoprecipitation (Co-IP) analysis shows the interaction between HtL1-HA and TaCDPK13-GFP. HtL1-HA and TaCDPK13-GFP were transiently co-expressed in *N. benthamiana* leaves. Immunoprecipitation was performed using anti-GFP magnetic beads. Three independent biological replicates were performed. **c** TaCDPK13 phosphorylates HtL1 in vitro. Phosphorylation modification of HtL1 was detected using anti-phosphoSer/Thr antibody. Three independent biological replicates were performed. **d** Vernalization inhibits phosphorylation of HtL1. Phosphorylation levels of constitutively overexpressed HtL1 were evaluated during various vernalization durations in *HtL1*-OE plants. Total proteins were extracted from plumules of wheat plants vernalized at 4 °C for 0, 7, 14 and 21 days, respectively. Phosphorylation modification of HtL1 was detected using anti-phosphoSer/Thr antibody, and

the constitutively overexpressed HtL1 was detected using anti-HA antibody. Actin was probed as a loading control. The phosphorylation levels were first normalized to HtL1, and then the relative phosphorylation levels at V7, V14, and V21 were further normalized to the level at V0 (set to 1). The relative protein abundance was quantified using ImageJ software. Three independent biological replicates were performed. **e** TaCDPK13 inhibits the aldolase activity of HtL1 in tobacco leaves. HtL1-HA was co-expressed with GFP and TaCDPK13-GFP, respectively, followed by immunoprecipitation of HtL1 using anti-HA antibody for enzyme activity assays. Three independent biological replicates were performed. Data are presented as mean ± SD, two-tailed Student's *t*-test for statistical analysis. **f** Vernalization significantly increases total aldolase activity in *HtL1*-OE plants. The plumules of the *HtL1*-OE3 line were vernalized at 4 °C for 0 and 21 days, and then collected for protein isolation and enzyme activity tests. Three independent biological replicates were performed. Data are presented as mean ± SD, two-tailed Student's *t*-test for statistical analysis. Source data are provided with this figure.

## TaOGT1-mediated *O*-GlcNAc modification of HtL1 enhances its stability

*O*-GlcNAcylation sites of HtL1 were previously identified through proteomics analysis of protein modifications[19], indicating a possible interaction of HtL1 with TaOGT1, an *O*-GlcNAc transferase reported to regulate flowering in wheat[18]. First, a yeast two-hybrid assay was performed, which revealed their interaction (Fig. 5a). Furthermore, BiFC, as well as LIC assays, confirmed this interaction in plants (Fig. 5b, c). Previous proteomic studies implied that *O*-GlcNAcylation of proteins plays a role in regulating vernalization in wheat[19]. Chemoenzymatic labeling and immunoblot analysis were used to detect the *O*-GlcNAc modification status of constitutively expressed HtL1 in *HtL1*-OE lines with or without vernalization. To examine the alterations in *O*-GlcNAc modification and the overall levels of HtL1 in response to vernalization, we subjected *HtL1*-OE plants to vernalization treatments with varying durations (V0, V7, V14, and V21). The results demonstrated a progressive accumulation of HtL1, accompanied by an increased proportion of *O*-GlcNAc modifications of HtL1 (Fig. 5d), following extended vernalization durations.

N-Acetylglucosamine (GlcNAc) can activate intracellular protein *O*-GlcNAc modification, and O-(2-acetamido-2-deoxy-D-glucopyranosylidene) amino-N-phenylcarbamate (PUGNAc), an inhibitor of *O*-GlcNAcase (OGA), enhances the levels of intracellular protein *O*-GlcNAcylation. Alloxan inhibits the activity of OGT, thereby reducing intracellular *O*-GlcNAc modification levels[19,44]. During germination, seeds of *HtL1*-OE plants were treated with combined GlcNAc and PUGNAc, as well as Alloxan alone, and then the constitutively expressed HtL1 in the plumules were detected with anti-HA antibody. Results showed that GlcNAc and PUGNAc together increased HtL1-HA protein levels (Fig. 5e), whereas plants treated with Alloxan showed a reduction in HtL1 protein levels (Fig. 5f). These results indicate that elevated overall *O*-GlcNAc modification levels resulted in an increase in HtL1 protein levels.

Taken together, these findings suggest that vernalization enhances the *O*-GlcNAc modification level of HtL1, leading to an elevated abundance of HtL1 by increasing its stability. Additionally, we generated overexpression lines of *TaOGT1* driven by the ubiquitin promoter in the KN199 background, and observed that *TaOGT1* overexpression plants exhibited early flowering in comparison to KN199 (Supplementary Fig. 16).

## HtL1 regulates flowering by mediating histone acetylation modifications at *VRN1*

Given the role of HtL1 in glycolysis, alterations in HtL1 activity may influence acetyl-CoA production by affecting the final product of the glycolytic pathway, which is related to histone acetylation. Immunoblotting analysis revealed that, compared to KN199, *HtL1*-RNAi lines

had significantly lower levels of H3K14ac and H3K27ac, but no significant change in H3K9ac abundance (Fig. 6a).

Given that *HtL1*-RNAi lines exhibited a strong delayed flowering phenotype and a significant reduction in mRNA abundance of *VRN1*, we further explored the correlation between *VRN1* transcription and the levles of H3K14ac and H3K27ac at the *VRN1* locus. Chromatin immunoprecipitation coupled with quantitative PCR (ChIP-qPCR) was performed to access the enrichment of H3K14ac and H3K27ac in the promoter and first intron regions of *VRN1* in A, B and D subgenomes in response to vernalization (Fig. 6b). We concurrently analyzed H3K4me3 levels in these regions. The H3K4me3 modification is enriched in the 5' promoter and gene body regions of flag leaves and spikelets in winter wheat following prolonged cold exposure, which is consistent with the sustained activation of *VRN1* during subsequent warm conditions following vernalization[9]. We found that under the conditions of a 21-day vernalization treatment, the H3K27ac, H3K14ac, and H3K4me3 levels in I, II, III, IV, V, and VI regions were all decreased in *HtL1*-RNAi plants compared to KN199 (Fig. 6c–e). To further confirm the relationship between HtL1-catalyzed reaction in the glycolytic pathway and flowering regulation, as well as histone acetylation at *VRN1* in response to vernalization, we evaluated the effects of exogenous application of FBP, the substrate for aldolase, on the flowering time of KN199 during the vernalization process. Phenotype observation showed that FBP-treated plants flowered earlier than the untreated control (supplementary Fig. 17a, b). In addition, the H3K14ac and H3K27ac levels at region IV were analyzed, revealing an elevation in both histone modifications (Supplementary Fig. 17c, d), and further RT-qPCR analysis confirmed increased transcription of *VRN1* under the same condition (Supplementary Fig. 17e). These findings collectively suggest that HtL1 promotes flowering through elevating H3K14ac and H3K27ac levels at *VRN1* to activate its transcription during vernalization process in wheat.

## Discussion
### *HtL1* regulates flowering in wheat
Vernalization-induced flowering time is a crucial agricultural trait, influenced by QTLs that impact grain yield. The quantitative trait of flowering time in wheat is regulated by a complex genetic network, in which *VRN1* plays a central role. Previous studies have identified major genes of QTLs that control vernalization response and flowering time traits[7,45–49], but less is known about how sugar metabolism, as a signaling pathway, genetically regulates vernalization for flowering. This study identified a significant SNP on chromosome 3A through GWAS, with *HtL1/FBA10* identified as a major gene influencing flowering. The geographical distribution and breeding selection analysis of *HtL1* revealed that the proportion of Hap-2 associated with longer flowering time gradually decreased in Chinese and American cultivars compared to landraces (Fig. 1g and Fig. 6f). It indicates a reduction in

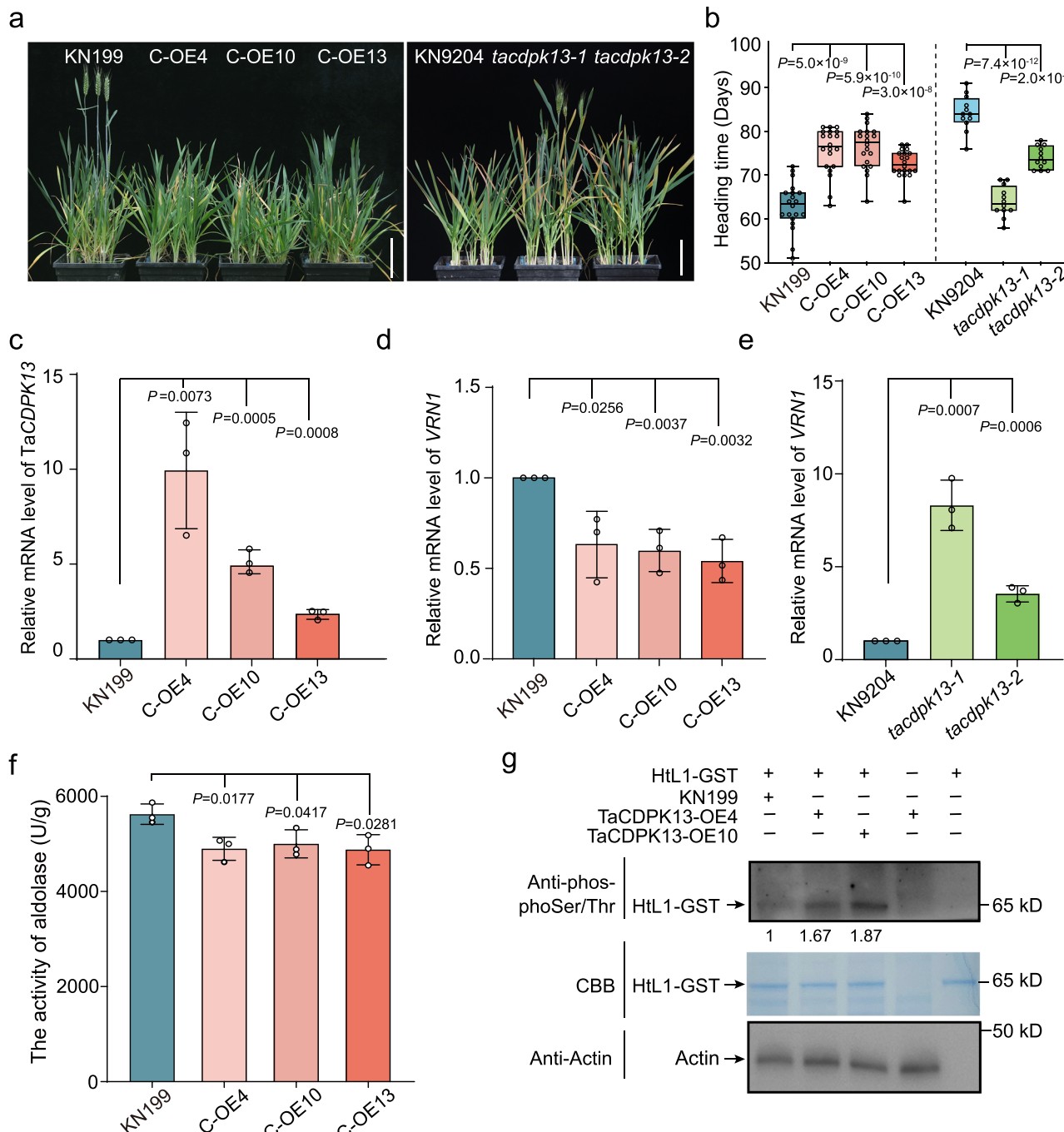

**Fig. 4 | *TaCDPK13* inhibits flowering in winter wheat. a** The morphological phenotype of *TaCDPK13* overexpression and *tacdpk13* mutant plants. *TaCDPK13*-OE plants were cultivated in the greenhouse following 21 days of vernalization, and photographs were captured 63 days after planting. The *tacdpk13* mutants were vernalized for 28 days, then moved to the greenhouse and photographed 61 days after planting. Scale bars, 10 cm. **b** Statistical analysis of heading time of *TaCDPK13*-OE lines and KN199, as well as *tacdpk13* mutants and wild-type KN9204. The box plots display the interquartile range, comprising the first quartile, median, and third quartile, while the whiskers extend from the minimum to the maximum values, (*n* = 20 plants in the *TaCDPK13*-OE group, and *n* = 12 plants for each line in the *tacdpk13* mutants group). Two-tailed Student's *t*-test for statistical analysis. **c** The relative mRNA abundance of *TaCDPK13* determined by RT-qPCR in *TaCDPK13*-OE and KN199 plants with 21 days of vernalization. The plants were vernalized for 21 days, following which RNA extraction was conducted. The transcript levels of *TaCDPK13* were normalized to *ACTIN*, then normalized to KN199 plants. Three independent biological replicates were performed. Data are mean ± SD, two-tailed Student's *t*-test for statistical analysis. **d** and **e** RT-qPCR analysis

shows the relative mRNA abundance of *VRN1* in *TaCDPK13*-OE lines and *tacdpk13* mutants, as well as their respective wild-type of KN199 and KN9204. *TaCDPK13*-OE lines and KN199 were vernalized for 21 days, *tacdpk13* mutants and KN9204 were vernalized for 28 days, following which RNA extraction was conducted. The transcript levels of *VRN1* were normalized to *ACTIN*, then normalized to wild-type plants. Three independent biological replicates were performed. Data are mean ± SD, two-tailed Student's *t*-test for statistical analysis. **f** The total aldolase activity analysis in *TaCDPK13*-OE plants with 21 days of vernalization. Three independent biological replicates were performed. Data are presented as mean ± SD, two-tailed Student's *t*-test for statistical analysis. **g** In vivo overexpressed TaCDPK13 increases the phosphorylation level of in vitro purified HtL1. Total proteins were isolated from plumules of *TaCDPK13*-OE lines and KN199 without vernalization, followed by incubation with purified HtL1-GST for subsequent phosphorylation analysis. Actin was probed as a loading control. Three independent biological replicates were performed. The relative protein abundance was quantified using ImageJ software. Source data are provided with this figure.

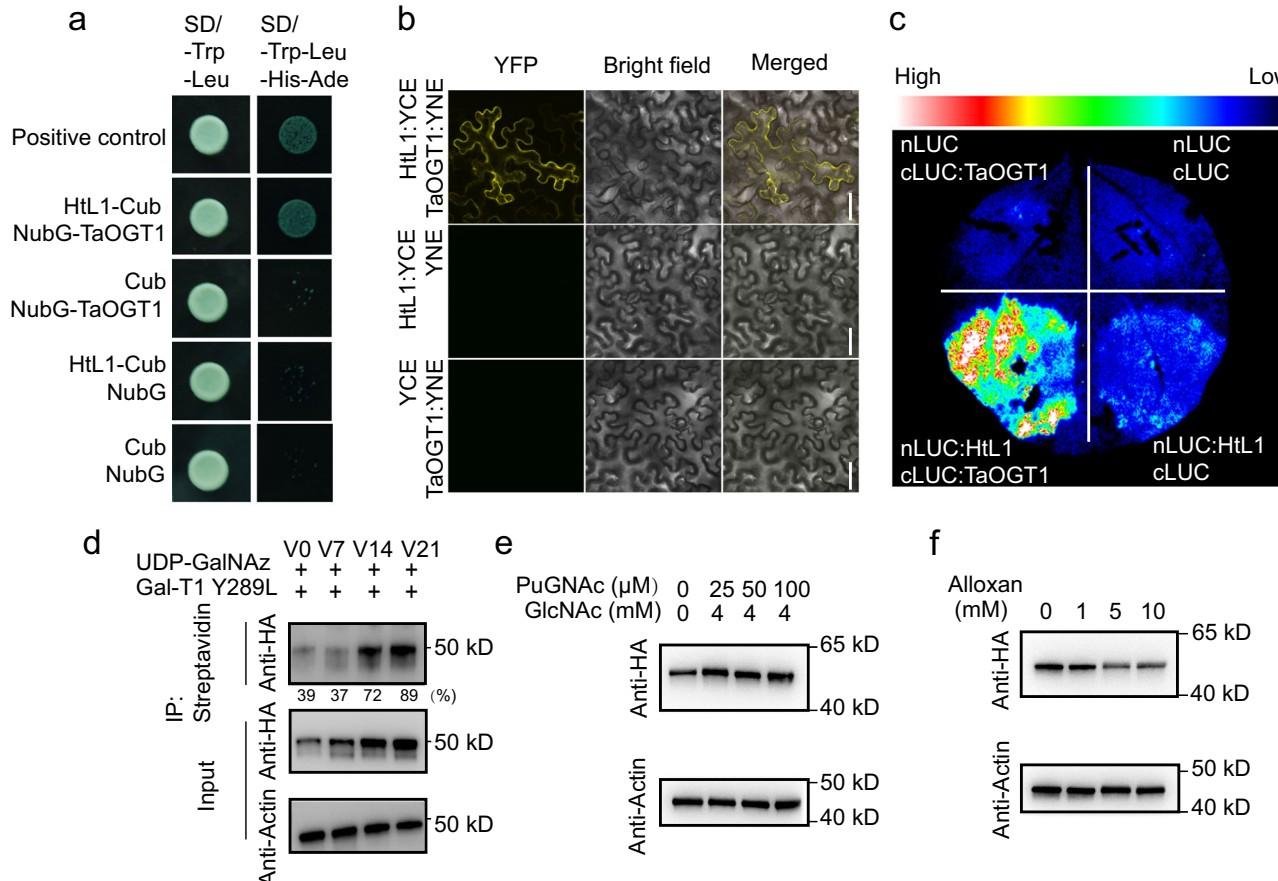

**Fig. 5 | O-GlcNAcylation of HtL1 is catalyzed by TaOGT1, and enhances its stability. a** Yeast two-hybrid (Y2H) assay shows the interaction between HtL1 and TaOGT1. APP-Cub and NubG-Fe65 was employed as the positive control. **b** Bimolecular Fluorescence Complementation (BiFC) assay indicates the interaction between HtL1 and TaOGT1 in vivo. Scale bars, 50 μm. **c** Detection of the interaction between HtL1 and TaOGT1 by Luciferase Complementation Imaging (LCI) assay. Luciferase activity was assessed 48 h after infiltration. **d** Vernalization induces an enhanced O-GlcNAcylation level of HtL1 and promotes its accumulation in wheat. Chemoenzymatic labeling approach was conducted for analyzing O-GlcNAcylated HtL1 in wheat. The *HtL1*-OE plants were vernalized at 4 °C for 0, 7, 14, and 21 days, respectively, and then for total protein isolation and chemoenzymatic labeling. O-GlcNAcylated proteins were selectively enriched for subsequent

detection of the target protein. Constitutively expressed HtL1 was detected using anti-HA antibody. The HtL1 levels were normalized to Actin, and then for quantification of the percentage of O-GlcNAcylated HtL1 relative its total amount. The relative protein abundance was quantified using ImageJ software. ($n = 3$ biological replicates for the analysis of HtL1-HA levels, and $n = 3$ biological replicates for chemoenzymatic labeling and the assay of O-GlcNAc modification levels of HtL1-HA following vernalization treatment). **e** and **f** Inhibitors that modify O-GlcNAc levels impact HtL1 stability. Germinating seeds of *HtL1*-OE plants were treated with PUGNAc combined with GlcNAc, or Alloxan alone during vernalization (V21), then for total protein extraction. HtL1 was detected using anti-HA antibody. Actin was probed as a loading control in (**d**), (**e**) and (**f**), respectively. Three independent biological replicates were performed. Source data are provided with this figure.

the breeding utilization of alleles associated with longer growth periods in both countries. This is consistent with previous studies showing a preference for favorable alleles associated with early flowering time in current wheat breeding in China and the United States[39]. Apart from Hap-2, distinct haplotypes are present in both countries, possibly influenced by varying environmental conditions and divergent human selection preferences, despite shared breeding objectives between the two countries. For instance, China and the United States employ different strategies when aiming to enhance production[39].

The regulatory role of carbohydrate in vernalization was established early in cereals. In particular, sucrose and glucose have stimulatory effects during the initial stages of vernalization[50]. However, the underlying molecular mechanism remains undisclosed. HtL1/FBA10, as a key enzyme with modifications, plays a crucial role in the glucose metabolism pathway. The *HtL1/FBA10* gene positively regulates flowering time during vernalization in wheat (Fig. 2a, b). FBAs play a key role in various abiotic stress responses, including salt and cold stress. In winter wheat, *FBA10* was found to enhance cold

tolerance by modulating glycolysis rate[51]. Calmodulin protein CML10 interacts with FBA6 through a Ca²⁺-dependent manner and enhances cold tolerance by activating the activity of FBA6 in *alfalfa*[52]. The promotion impact of *HtL1/FBA10* on vernalization underscores its significance in enabling winter wheat to thrive in prolonged cold conditions, thereby ensuring normal growth and development. These discoveries prompt further exploration of the interplay between phosphorylation and O-GlcNAc modifications in regulating the activity of HtL1/FBA10, a crucial enzyme in the glucose metabolism pathway, to unravel the functional mechanism governing vernalization response in wheat.

## Phosphorylation and *O*-GlcNAc modifications regulate the function of HtL1

As known, phosphorylation plays a crucial role in regulating enzyme activity and protein degradation during low-temperature signal transduction in plants, influencing cold tolerance, growth, and development[53]. In wheat, glycogen synthase kinase 3 (GSK3), a

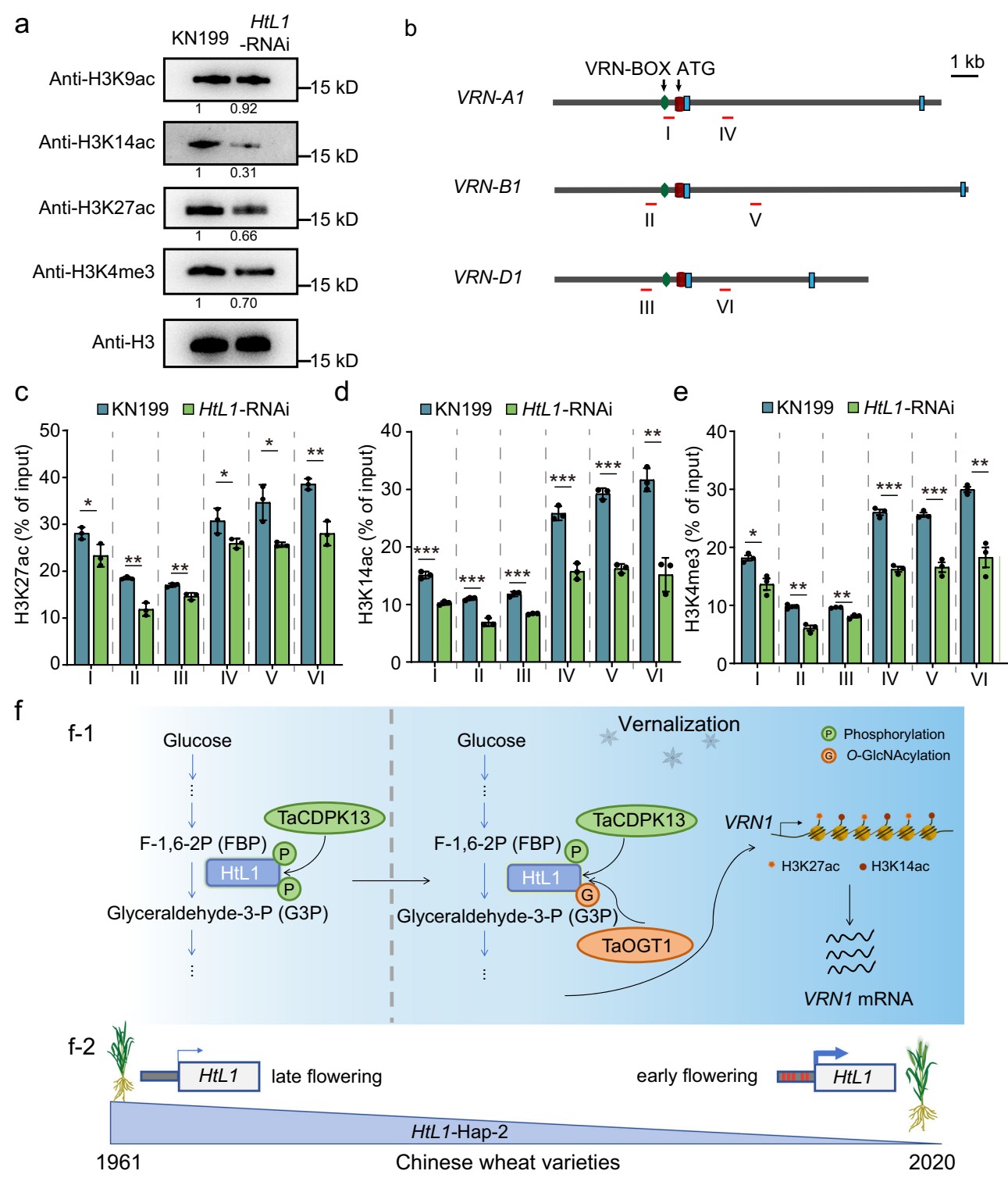

serine/threonine protein kinase, regulates flowering by phosphorylating the key vernalization response protein VRN1[54]. In this study, the novel functional mechanism of HtL1/FBA10 involves a trade-off between phosphorylation and *O*-GlcNAcylation to regulate vernalization-induced flowering. Vernalization treatment decreases phosphorylation levels of HtL1 while increasing its *O*-GlcNAcylation levels (Figs. 3d, 5d). Interestingly, these two modifications exhibit opposing trends during vernalization, suggesting that phosphorylation and *O*-GlcNAc modification jointly regulate the function of HtL1 in response to prolonged cold exposure (Fig. 6f). Both *O*-GlcNAcylation and phosphorylation patterns of proteins are involved in human diseases. For instance, multiple amino acid residues on the insulin receptor substrate IRS-1 undergo *O*-GlcNAc modification, with adjacent sites also subject to phosphorylation[55]. Inhibition of phosphorylation leads to changes in *O*-GlcNAc modification levels on proteins[56]. In addition, GSK3 kinase inhibitors enhance *O*-GlcNAcylation of specific proteins in cells[57]. Nevertheless, the understanding of the interplay between phosphorylation and *O*-GlcNAc modifications of target proteins in plants during developmental processes remains limited.

**Fig. 6 | Loss of function of *HtL1* influences histone modification states at the *VRN1* locus. a** Detection of overall levels of H3K9ac, H3K14ac, H3K27ac, and H3K4me3 modification in *HtL1*-RNAi and KN199 plants vernalized for 21 days. Three biological replicates were performed. Histone H3 was probed as a loading control, and the levels of histone modifications were normalized to H3. **b** Schematic diagram of *VRN1*, showing fragments detected in ChIP-qPCR analysis. Exons are represented by blue boxes, and the regions examined by ChIP are indicated by numbered red bars. ChIP-qPCR assays of H3K27ac (**c**), H3K14ac (**d**) and H3K4me3 (**e**) levels in the indicated regions at *VRN1* in *HtL1*-RNAi and KN199 plants following 21 days of vernalization. Three biological replicates were performed. Data are mean ± SD, *P < 0.05, **P < 0.01, ***P < 0.001 by two-tailed Student's *t*-test. The relative abundance was normalized to the input. **f** A working model for understanding the role of *HtL1* in vernalization-mediated flowering and breeding selection analysis of *HtL1*. (f-1) The activity of fructose 1,6-biphosphate aldolase is regulated by phosphorylation and *O*-GlcNAc modifications during vernalization. Before vernalization, HtL1 was phosphorylated by TaCDPK13, and its aldolase activity was inhibited. During the vernalization process, the phosphorylation modification level of HtL1 gradually decreased, while *O*-GlcNAc modification of HtL1 was enhanced, resulting in increased activity and stability of HtL1. The increased activity of aldolase HtL1 promotes the enrichment of H3K14ac and H3K27ac at the *VRN1* locus, leading to the activation of *VRN1* transcription and consequently accelerating flowering in wheat. (f-2) Wheat varieties possessing variations in the *HtL1* promoter regions exhibit differences in flowering time. The red vertical lines in the promoter region denote single-nucleotide polymorphisms (SNPs) at these sites in these varieties relative to the Chinese Spring reference genome. The utilization of wheat varieties with the Hap-2, which is associated with delayed flowering, has gradually decreased during the modern breeding process in China. Source data are provided with this figure.

Our study identified TaCDPK13 as a calcium-dependent protein kinase that phosphorylates HtL1. This phosphorylation leads to the inhibition of HtL1 activity (Fig. 3e). Notably, the phosphorylation levels of HtL1 decrease following vernalization treatment (Fig. 3d). Vernalization treatment significantly increased *O*-GlcNAc modification and the protein levels of HtL1 (Fig. 5d), suggesting that *O*-GlcNAc modification could enhance the stability of HtL1. The flowering phenotypes of transgenic wheat lines overexpressing *TaCDPK13* and *TaOGT1* are consistent with a role for these enzymes in the post-translational regulation of HtL1 (Fig. 4a and Supplementary Fig. 16a). Furthermore, the treatment of seedlings with inhibitors targeting calcium-dependent protein kinases resulted in an overall enhancement in protein *O*-GlcNAc modification levels (Supplementary Fig. 18). It is hypothesized that during vernalization, *O*-GlcNAcylation antagonize calcium-dependent phosphorylation on HtL1, thereby enhancing its aldolase activity. These results highlight the regulatory role of HtL1 in response to vernalization in winter wheat.

### HtL1 epigenetically regulates vernalization response in wheat

The key regulatory genes *VRN1*, *VRN2*, and *VRN3* form a network that orchestrates the vernalization response for flowering in cereals[41,58,59]. In contrast to the vernalization-induced epigenetic silencing of the flowering repressor *FLC* in *Arabidopsis*, prolonged cold exposure results in the epigenetic activation of the flowering promoter *VRN1* in temperate grasses[10,41]. In winter wheat, vernalization-induced activation of *VRN1* involves active histone marks of H3K4me3 and H3K36me3, which together with H3K27me3 play crucial roles in vernalization response and embryonic resetting[9].

In this study, we demonstrated that genetic modifications of *HtL1* alter *VRN1* transcription during vernalization, and knockdown of *HtL1* decreases transcription levels of *VRN1* and reduces accumulation of H3K14ac and H3K27ac in the promoter and first intron regions at *VRN1* (Fig. 2d and Fig. 6c, d). These results suggest that HtL1 enhances *VRN1* transcription to promote flowering by elevating histone acetylation modifications at *VRN1* during vernalization process. This implies that fructose-1,6-biphosphate aldolase may serve as a connection between glucose metabolism and epigenetic modifications. Additionally, *HtL1*-RNAi plants show a decreasing trend in H3K4me3 modification in both the promoter and first intron regions of *VRN1* in comparison with the wild type (Fig. 6e). In *Arabidopsis*, the MRG proteins functions as a "reader" for H3K36me3 and recruits a H4-specific histone acetyltransferase, indicating that H3K36me3 and histone H4 acetylation interplay at target loci associated with flowering[60]. The roles of H3K14ac and H3K27ac in the activation of *VRN1* during vernalization are less known in wheat. Therefore, we propose the hypothesis that HtL1-mediated H3K14ac and H3K27ac modifications may synergistically interact with H3K4me3 to activate *VRN1* during vernalization, which requires further experimental validation. In summary, our findings reveal a previously unrecognized mechanism of aldolase in the regulation of plant development and environmental adaptability.

## Methods

### Plant materials and growth conditions

Wheat (*Triticum aestivum* L.) cultivar KN199 was used for gene cloning and genetic transformation. Wheat plants were grown at a diurnal temperature regime of 22 °C during the day and 18 °C at night under long-day conditions (16-h light/8-h dark) in an illuminated plant growth chamber, with a relative humidity of 65%. The seeds were surface-sterilized in 2% NaClO for 20 min, followed by thorough rinsing with flowing water. After imbibition, the seeds were placed on moistened filter papers and allowed to germinate in darkness at 25 °C for 3 days, and then were subsequently subjected to a vernalization treatment at 4 °C for predetermined durations in complete darkness. Seeds that were germinated and cultivated at 25 °C for three days served as the non-vernalized control group (V0). For phenotype observation and tissue harvesting, vernalized or non-vernalized seedlings were transplanted into soil and cultivated in an environmentally controlled greenhouse. Fresh samples were harvested and rapidly frozen in liquid nitrogen and stored at −80 °C. Heading time, defined as the date when the primary spike fully emerged from the flag leaf sheath, was statistically analyzed in this study.

### GWAS analysis

The GWAS was performed for heading time traits on 91 wheat accessions by GEMMA (Genome-wide Efficient Mixed Model Association) software according to methods previously described[37,61], and using the mixed linear model (MLM) analysis.

$$y = X\alpha + S\beta + K\mu + e$$

In this model, y represents phenotype trait; α and β for fixed effects, corresponding to marker effects and non-marker effects respectively; while μ represents unknown random effects. X, S, and K serve as the incidence matrices associated with α, β, and μ, and e is the vector of random residual effects. For constructing the S matrix, the first three principal components (PCs) were utilized through Plink software (version 1.90b6.18). The kinship matrix (K) was established based on the simple matching coefficient matrix. Both the S matrix and the kinship matrix (K) were employed to account for population stratification, thereby correcting for potential population structure effects.

Significant *P* value thresholds ($P < 10^{-5}$) were set to control the genome-wide type I error rate. Then we enlarged the candidate region to 1000 kb centered on the GWAS signal peak to identity candidate genes. The GWAS results were visualized using the R software package CMplot[62], which included the creation of QQ plots and Manhattan

plots. Additionally, linkage disequilibrium analysis was performed using the PopLDdecay software[63].

## Generation of transgenic wheat materials

The full-length coding sequences of *HtL1*, *TaCDPK13*, and *TaOGT1* were effectively amplified from the winter wheat cultivar KN199. These sequences were then recombined into the *p*UN1301 vector, which contains the maize ubiquitin promoter to allow constitutive expression of HtL1, TaCDPK13, and TaOGT1, each fused with HA and FLAG tags, respectively.

For the generation of CRISPR/Cas9-mediated *FBA10/HtL1* knockout mutants, single guide RNAs (sgRNAs) targeting the second exon were designed using the E-CRISP Design website (http://www.e-crisp.org/E-CRISP.html). Subsequently, the target site sequences were cloned into a CRISPR/Cas9 vector linearized with *BsaI* (Thermo Fisher Scientific, ER0292) according to the manufacturer's instructions. RNAi was used to generate *HtL1* knockdown mutants. The RNAi vector was constructed to target the second exon of *HtL1*, a region selected for its minimal sequence identity to other homologous aldolase-encoding genes. All constructs were introduced into *Agrobacterium tumefaciens* strain EHA105 and then transformed into KN199.

## RNA extraction, reverse transcription and RT−qPCR

Total RNA was isolated using the TRIzol reagent (Thermo Fisher Scientific, 15596026), and reverse transcription was performed with the HiScript II Q RT SuperMix by following the manufacturer's protocol (Vazyme, R223-01). Real-time fluorescence quantitative PCR (RT-qPCR) was conducted on diluted cDNA from three biological replicates using SYBR qPCR Master Mix (Vazyme, Q712-02) in an Mx3000P system. Wheat *ACTIN* was used to normalize the gene expression levels. The primer sequences for RT-qPCR can be found in Supplementary Data 6.

## Library preparation and RNA-Sequencing

Total RNA was isolated using TRIzol Reagent (Thermo Fisher Scientific, 15596026) following standard protocols. Genomic DNA contamination was eliminated through DNase I digestion (extremely England Biolabs, M0303L). RNA purity was verified by measuring the A260/A280 ratio (NanoDrop™ OneC, Thermo Fisher Scientific), and RNA integrity was verified via the LabChip GX Touch system (Revvity). RNA concentration was quantified fluorometrically using a Qubit 3.0 system with the Qubit RNA Broad Range Assay kit (Thermo Fisher Scientific, Q10210). mRNA libraries were constructed from 1 μg total RNA using the KC™ Digital mRNA Library Prep Kit (SeqHealth Technology Co., Ltd., Wuhan, China) following the manufacturer's protocol. To mitigate PCR amplification bias and sequencing errors, the kit incorporates 12-nucleotide unique molecular identifiers (UMIs) for precise labeling of cDNA molecules during first-strand synthesis. Size selection targeting 200–500 bp fragments was performed using magnetic bead-based purification. The enriched libraries were quantified and subjected to PE150 paired-end sequencing on the DNBSEQ-T7 platform (MGI).

## Chromatin immunoprecipitation (ChIP)

ChIP assays were conducted following previously established protocols[64]. Briefly, Total chromatin was fragmented via sonication and subsequently subjected to immunoprecipitation using antibodies against H3K27ac (Abcam, ab4729), H3K14ac (Abcam, Ab52946), or H3K4me3 (Millipore, 07473). The quantification of immunoprecipitated *VRN1* chromatin was conducted utilizing qPCR, focusing on the specified regions within the *VRN1* locus. The relative abundances of histone modifications were normalized to the input DNA. The primer sequences were provided in Supplementary Data 6.

## Immunoblot analysis and Co-IP

Total proteins were extracted from samples and quantified by Bradford assay. The protein samples were boiled at 95 °C for 10 min,

centrifuged for 10 min, then were separated by 10% SDS-PAGE gels and transferred onto polyvinylidene fluoride membrane (Bio-Rad). The membrane was blocked using 5% (W/V) defatted milk or 5% (W/V) BSA (Bovine Serum Albumin) in 1×Tris-buffered saline containing Tween 20 (TBST, 1 × TBS with 0.05% [v/v] Tween-20) at room temperature for 2 h or overnight at 4 °C. Subsequently, it was incubated with the primary antibody (anti-Actin [Huaxingbio, HX1843], anti-H3 [Abcam, ab1791], anti-H3K9ac [Abcam, ab218553], anti-CTD110.6 [Cell Signaling Technology, 9875S]) at room temperature for 2 h or at 4 °C overnight. Following this, the membrane underwent three washes with 1 × TBST for 10 min each time. It was then incubated with an HRP-conjugated secondary antibody (1:5000 dilution) for 1 h at room temperature, followed by three washes with 1 × TBST. The chemiluminescence signals were detected using a Tanon-5200 system (Tanon). The band intensities on Western blot images were quantified using ImageJ software.

For Co-IP experiments, the full-length coding sequences of *HtL1* and *TaCDPK13* were inserted into *p*CsVMV-HA3-N-1300 and *p*BI121-GFP vectors, respectively, and then co-transformed into *Agrobacterium tumefaciens* strain GV3101 and infiltrated into *N. benthamiana* leaves. After 72 h, total proteins were extracted from the transfected leaves and incubated with anti-GFP magnetic beads (Chromotek, gtma-100) at 4 °C for 4 h. Immunoprecipitated proteins were subsequently detected through immunoblot analysis.

## Yeast two-hybrid assay

For yeast two-hybrid analysis, the full-length coding sequence of the *HtL1* gene was cloned into the *p*BT3-N vector, while the entire coding region of *TaOGT1* was inserted into the *p*PR3-N vector. To assess protein interactions in yeast, the prey plasmid containing the target gene was co-transformed with the bait construct into the yeast strain MY51 according to the manufacturer's manual (Clontech, USA). The transformed yeast cells were subsequently cultured on SD medium lacking Trp and Leu (SD/-Trp/-Leu) at 30 °C for 2 days. Subsequently, positive transformants were screened on SD/-Trp/-Leu/-His-Ade (20 μg/mL X-gal, 50 mM 3-AT).

## Firefly Luciferase Complementation Imaging (LCI) assay

The LCI assay was conducted to detect protein interactions between HtL1 and TaCDPK13 or TaOGT1 in *N. benthamiana*. Briefly, the full-length coding sequences of *HtL1, TaOGT1* and *TaCDPK13* were inserted into *p*CAMBIA1300-cLuc or *p*CAMBIA1300-nLuc vectors separately. *Agrobacterium* strain GV3101 containing the specified constructs were co-infiltrated into *N. benthamiana* leaves. The luminescence images reflecting Luc activity were captured using a CCD imaging system.

## BiFC and subcellular localization assays

The full-length coding sequences of *HtL1*, *TaCDPK13* and *TaOGT1* were inserted into the *p*UC-SPYCE and *p*UC-SPYNE vectors, respectively. Different combinations of constructs were then co-transformed into *N. benthamiana* leaves. Confocal microscopy (Leica TCS SP5) was employed to observe the fluorescence signals of YFP.

For the HtL1 subcellular localization assay, the coding sequence of *HtL1* was cloned into *p*BI221-GFP and *p*BI121-GFP vectors, respectively, allowing the expression of HtL1-GFP fusion protein. The control and fusion constructs were independently introduced into *Arabidopsis* protoplasts by a polyethylene glycol-mediated transient expression system, or infiltrated into *N. benthamiana* leaves via *Agrobacterium* (strain GV3101)-mediated transformation. Confocal microscopy (Leica TCS SP5) was employed to observe the fluorescence signals of GFP.

## Aldolase activity assay

The full-length coding sequences of *HtL1* and *TaCDPK13* were individually inserted into *p*CsVMV-HA3-N-1300 and *p*BI121-GFP vectors, respectively, and then co-transfected into *N. benthamiana* leaves. The

combinations HtL1-HA and GFP were used as controls. To purify HtL1-HA, total protein was extracted 48 h after transfection using a lysis buffer. The protein sample was incubated with anti-HA magnetic beads (Pierce, 88837) at 4 °C overnight, then the HtL1-HA protein was eluted with HA peptide (Pierce, 26184) according to the manufacture's protocol. The eluent was further purified and concentrated using an Amicon Ultra Centrifugal Filter (Millipore). Aldolase activity of purified HtL1-HA or cell lysates prepared from wheat plumule was determined using a fructose-1,6-bisphosphate aldolase activity assay kit (Solarbio, BC2275) with a microplate reader.

### In vitro and in vivo phosphorylation assays

To detect phosphorylation modification of HtL1, the purified recombinant HtL1-GST protein was used as a substrate and incubated with TaCDPK13-GST in a kinase reaction buffer (25 mM Tris-HCl pH 7.5, 5 mM β-Glycerophosphoric acid, 2 mM DL-Dithiothreitol, 0.1 mM $Na_3VO_4$, 10 mM $MgCl_2$) at 30 °C for 40 min. The reaction was terminated by adding 5×SDS loading buffer and boiling at 95 °C for 10 min. A portion of phosphorylated protein was separated by 10% SDS-PAGE and detected using an anti-phosphoSer/Thr antibody[65] (1:2000 dilution). The gel stained with Coomassie brilliant blue (CBB) was used as a loading control.

Total protein was extracted from *HtL1*-OE plants. Whereafter, immunoprecipitation of HtL1 was performed using anti-HA magnetic beads (Pierce, 88837), followed by detection of the samples through 10% SDS-PAGE and immunoblotting with anti-phosSer/Thr antibody (1:2000 dilution) and anti-HA antibody (Sigma, H9658,1:5000 dilution).

### LC-MS/MS analysis

The fusion proteins HtL1-GST and TaCDPK13-GST, expressed in *E. coli* strain BL21 (DE3), were purified and incubated in a kinase reaction buffer (25 mM Tris-HCl pH 7.5, 5 mM β-glycerophosphoric acid, 2 mM DL-dithiothreitol, 0.1 mM $Na_3VO_4$, 10 mM $MgCl_2$). After incubation at 30 °C for 40 min, the proteins were subjected to digestion. Specifically, the disulfide bonds were first denatured and reduced by incubation with 5 mM TCEP at 55 °C for 30 min. Subsequently, the samples were treated with 25 mM iodoacetamide and incubated in the dark at room temperature for 30 min. Finally, the proteins were digested with approximately 1 μg of sequencing-grade trypsin at 37 °C for 16 h, and the resulting samples were analyzed by mass spectrometry. For a detailed experimental procedure, please refer to the supplementary method included in the supplementary information file.

### Enzymatic labeling analysis of HtL1 *O*-GlcNAcylation

Proteins containing *O*-GlcNAc residues in cell lysates were labeled using enzymatic labeling and biotinylation following the manufacturer's protocol (Thermo Fisher Scientific, C33368). In brief, total protein was extracted in lysis buffer (20 mM Tris-HCl, pH 7.5, 150 mM NaCl, 10 μM PUGNAc, Protease inhibitor cocktail), and the Click-iT *O*-GlcNAc Enzymatic Labeling System was utilized to label the cell lysate at 4 °C for 24 h. Subsequently, the sample was reacted with biotin alkyne after removing excess UDP-GalNAz, according to the manufacturer's protocol of the Click-iT Protein Analysis Detection Kit (Thermo Fisher Scientific, C33372). Biotin-labeled samples were then incubated with Streptavidin Magnetic Beads (Beyotime, P2151) overnight at 4 °C. Afterwards, anti-HA antibody (Sigma, H9658, 1:5000 dilution) was employed to detect the levels of *O*-GlcNAc modified HtL1.

### Statistics and reproducibility

Data were analyzed using GraphPad Prism software (version 10.4). For comparisons between two groups, an unpaired two-tailed Student's *t*-test was applied, while one-way ANOVA was used for comparisons across multiple groups. Significant differences are denoted by asterisks (*$P < 0.05$, **$P < 0.01$, ***$P < 0.001$) or by different lower-case letters ($P < 0.05$). All Western blot experiments were independently repeated three times, with consistent results obtained across replicates. The sample size was not predetermined statistically, and no data were excluded from the analyses. The study was conducted without randomization or blinding of the investigators to group allocation during the experiments and outcome assessment.

### Reporting summary

Further information on research design is available in the Nature Portfolio Reporting Summary linked to this article.

## Data availability

The raw RNA-sequence data generated in this study have been deposited in the Genome Sequence Archive[66] in National Genomics Data Center[67], China National Center for Bioinformation/Beijing Institute of Genomics, Chinese Academy of Sciences (https://ngdc.cncb.ac.cn/gsa) under accession code CRA028443. The raw mass spectrometry data for protein phosphorylation residue identification generated in this study have been deposited in the ProteomeXchange Consortium via the PRIDE[68] partner repository under accession code PXD069370. Additional relevant materials can be obtained from the corresponding author upon request. Source data are provided with this paper.

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

## Acknowledgements

This work was supported by the Basic Science Center Project of National Natural Science Foundation of China (32388201) to K.C and the National Natural Science Foundation of China (31970331) to L.X. We thank Dr. Zhuang Lu, Dr. Bin Han and Ms. Jingquan Li (Plant Science Facility of the Institute of Botany, Chinese Academy of Sciences) for their technical assistance in LC-MS/MS assay, small molecule compound analysis and the subcellular localization assay, respectively. We thank Dr. Wei Luo and Dr. Dongfeng Liu for helpful discussions.

## Author contributions

K.C. conceived the project. K.C., L.X., Z.G. and Y.X. designed the research. P.Y., Y.L., Q.D., Y.M. and Y.N. conducted the experiments. J.Z., S.X., H.Z., X.Z. and Y.X. analyzed the data. P.Y., Y.L., Z.G., L.X. and K.C. wrote and revised the paper. All authors read and approved the final manuscript.

## Competing interests

The authors declare no competing interests.
