## [Transparent Peer Review file · Nature Communications]

O-GlcNAc and phosphorylation modifications on HtL1/FBA10 regulate wheat vernalization for flowering

Corresponding Author: Professor Kang Chong

Version 0:

Reviewer comments:

Reviewer #1

(Remarks to the Author)

This manuscript studies the vernalization response in wheat. It starts with a genetic analysis and identifies a QTL (qHTL1) associated with heading time after vernalization. They identified the causal gene of qHTL1 as FBA10 that encodes the enzyme fructose 1,6-bisphosphate aldolase. Overexpression of FBA10 caused early flowering whereas knock down caused late flowering, and also either increased or decreased aldolase activity. This correlates with increased or reduced levels of VRN1 mRNA, a promoter of flowering after vernalization. They then argue that post-translational modification of FBA10 is altered during vernalization. Phosphorylation of FBA10 is reduced during vernalization whereas O-GlcNAcylation is increased. These changes are proposed to increase FBA10 protein stability and lead to changes in histone modification at the VRN1 locus that change its expression.

This manuscript contains a lot of work going all the way from genetic analysis to detailed biochemical and genomic experiments. However, I was not convinced by several of the conclusions and judged that the data shown were insufficient. I realize the difficulties of working in hexaploid wheat, but perhaps a more focused article that made fewer points but examined them more closely would be more convincing. A smaller point is that the writing needs detailed, perhaps professional, editing in many places.

Detailed comments:

1. Resolution of the identification of FBA10. The genetic analysis identifies a large interval including at least 22 genes. The logic for then focusing on FBA10 was suggestive but not totally convincing based on expression level and that the protein is modified during vernalization (lines 122 – 124). Here I wondered what is the difference in sequence between FBA10 alleles, and how does this correlate with gene activity? Also, how many other genes encode FBA10 enzymes in the wheat genome, and how would genetic modifications at this gene affect the total pool of enzyme? Often metabolic enzymes are encoded by genes at multiple loci, making it less clear why modification of a single copy would impact phenotype.
2. The genetic testing of FBA10 is rather crude (lines 127-129). This is based on very strong overexpression or knock down by RNAi. The knock down could affect other genes as well as the gene of interest and this does not seem to have been tested. It should be possible now even in wheat to make CRISPR alleles of specific genes.
3. I wondered about the pleiotropy of FBA10 knock down. Since this is an enzyme involved in primary metabolism, do the knock down plants grow more slowly and does this explain at least partly the delay in heading date?
4. Is the effect on phenotype expected from the effect on gene expression? In Figure 1b-d. I found it surprising that in the OE lines VRN1 expression was increased 4-5x but the heading time phenotype was very small.
5. I did not find the conclusion that protein modification affected FBA10 protein stability convincing, because I would have liked to see mRNA analysis so that any transcriptional effect could be assessed. Also, I would have liked to see levels of FBA10 protein relative to modified protein assessed in each of the experiments. In Figure 5g, FBA10 protein levels seem to increase during vernalization, but does the level of FBA10 mRNA or is this increase purely due to enhanced stability? In Figure 3d, the level of phosphorylated protein decreases during vernalization, but in this experiment does the level of protein increase as in 5g or not? I would like to see the level of phosphorylated protein relative to the total amount of FBA10 in the same extracts. Same for other modifications.
6. A small point is that there are many words and phrases requiring editing. In the first two lines of the abstract vernalization is spelt wrongly, and “remains blurry” is not really scientific. In line 32 “involved in two natural hybridization” at least lacks a plural. But there are many examples.

Reviewer #2

(Remarks to the Author)

This study describes a candidate gene identified by GWAS on chromosome of 3A from wheat that potentially underlies variation in heading date (the timing of flowering) amongst a collection of predominantly Chinese wheat varieties – qHtL1.

The authors propose that a fructose 1,6-diphosphate aldolase 10 is the gene underlying the marker trait associations observed in the region of interest (qHtL1). Then, through biochemical characterisations and analyses of heading date and molecular phenotypes of transgenic plants, the authors provide evidence that biochemical interactions between the candidate gene and a kinase (CDPK13) lead to histone modifications of the VRN1 gene that influence heading date.

This is an interesting and worthwhile topic. The authors present a lot of biochemical data to build a mechanistic model to explain at a molecular level the mode of action for qHtL1.

Overall the manuscript is clear. There are a number of areas where more information would strengthen the manuscript and support the overall conclusions.

The most important point to address is the selection of the candidate gene. At the moment the information used to select the candidate gene was (1) expression data, and (2) statistical association. The weakness here is that all genes in that interval will have a statistical association – the GWAS data tells us that, the genes are all closely linked and will share a common statistical association with the heading date trait. So, the statistical evidence creates circular logic in the current version of the manuscript.

Noting also that one other gene in the genetic interval appears to be a known regulator of histone modifications of MADs box genes? (TraesCS3A02G391200 is related to FLX). It would be good to rule out this gene as a candidate.

A suggestion is to provide a table summarising sequence expression patterns and natural variation identified in all the candidate genes in that interval.

Alternatively, and even more powerful, would be to isolate loss-of-function mutants in the prime candidate gene to provide causal evidence that this gene controls flowering, beyond the transgenic plants used in this study.

1. The abstract could be improved.
2. More information is needed about the source of heading-date data used for GWAS. What were the locations and conditions used for these experiments?
3. It was hard to evaluate the importance of qHtL1 relative to overall variation in heading date. A more comprehensive Manhattan plot should be presented to show the complete picture.
4. Similarly, there needs to be more information provided to support the GWAS analysis. What p-value cut-offs were applied? Was correction of population structure applied?
5. Line 142. More details need to be provided to show how the connection between qHtL1 and CDPK13 was revealed? The method and the data generated are not explained
6. Please check the manuscript for minor errors. For example, in a few instances there are), at ends of sentences, instead of).

I wish the authors the best of luck with this research.

Reviewer #3

(Remarks to the Author)

Yang et al. aim to dissect the mechanisms regulating vernalization for flowering in wheat. Using a genome-wide association study (GWAS), they mapped an important gene associated with flowering time, qHtL1, which encodes fructose 1,6-bisphosphate aldolase (FBA). They showed that overexpression of this gene increased aldolase activity, while RNAi knockdown decreased it, confirming that FBA does indeed have aldolase activity. The overexpression lines showed slightly earlier flowering, while the knockout lines showed delayed flowering, positioning FBA as a key candidate in this region for regulating wheat vernalization using genetic tools.

The authors further investigated the regulation of FBA and demonstrated its modulation by CDPK13, a kinase, and TaOGT, which mediates O-GlcNAcylation. Their biochemical data suggest that vernalization decreases the phosphorylation level of FBA, which in turn increases its activity. Overexpression of CDPK13 delayed flowering, while its mutation resulted in an earlier flowering phenotype. In addition, they showed that FBA undergoes O-GlcNAc modification in vivo, with FBA protein levels increasing upon vernalization. Notably, loss of FBA protein affects the state of histone modification on VRN1 chromatin, thereby regulating VRN1 expression.

Overall, the paper is interesting. However, several figures have conflicting results and some data need further clarification and quality improvement.

In Figure 3d, qHtL1-HA protein levels appear to be consistent across the V0, V7, V14, and V21 stages. However, Figure 5g shows conflicting results, indicating higher qHtL1 protein levels at the V21 stage compared to V0. It is unclear which figure accurately reflects the true levels of qHtL1 and this discrepancy needs to be resolved for consistency.

The authors argue that CDPK13 regulates qHtL1, and Figure 3d shows decreased phosphorylation levels using an antibody in a Western blot. It is crucial to identify the exact modification sites in vivo that show this decrease and to confirm whether these sites correspond to those identified in vitro by incubation of TaCDPK13 with qHtL1. If the sites overlap, this would

provide stronger evidence that CDPK13 directly modifies qHtL1 and plays a key role in regulating its phosphorylation, thereby affecting qHtL1 activity. A remaining question, however, is how CDPK13 itself is regulated by vernalization. Figure 4g shows an increase in phosphorylation when purified qHtL1-GST is incubated with TaCDPK13-OE4 cell lysate, indicating a slight increase in qHtL1 phosphorylation. However, the quality of the data is insufficient. Western blot quantification is not convincing and requires multiple replicates for validation. In addition, quantification of phosphorylation by alternative methods would increase the reliability of the results.

Figure 5 illustrates the regulation of qHtL1 by TaOGT1; however, it is unclear how many replicates were performed to ensure the reproducibility of the data. Furthermore, there seems to be an inconsistency in the interpretation of the data between Figure 5g and Figure 5d. If Figure 5g shows a multi-fold increase in qHtL1-HA protein levels following vernalization treatment, how does this correlate with the consistent input levels shown in Figure 5d? This discrepancy raises questions about the experimental design and normalization methods used. In addition, the number of replicates performed and the reproducibility of these results are not clearly stated, which needs to be addressed to increase confidence in the results. If Figure 5g shows that the qHtL1 level increases several fold between v0 and v21, it is puzzling why the activity of the aldolase level increases only very slightly (900 to 1200?).

To prove the model, is it possible to treat the wheat with the product (G3P) of qHtL1 and show that it would affect the histone modification states on VRN1 and affect the flowering time?

Additional Points:

All figures from Figure 1 through Figure 6 use traditional bar graphs. To improve visual clarity and allow for a better understanding of the actual data distribution, box plots or beeswarm plots should be used.

Version 1:

Reviewer comments:

Reviewer #1

(Remarks to the Author)

I appreciate that the authors have carefully considered my comments on the previous version and have provided a thoroughly argued response that includes new data.

However, I remain unconvinced that they have proven that TraesCS3A02G391100/FBA10/qHtL1 is the causal gene for the heading date phenotype. For me, key issues are to do with the functional differences between the haplotypes and the genetic confirmation that this is the causal gene. The genetic confirmation is complicated by the result they mention in the response document that a CRISPR-induced allele is lethal – so they cannot test its effect on heading date. I think that they should include this CRISPR allele and the lethality phenotype in the manuscript, so that the reader knows that this experiment was done. In the response document, they go on to say “In contrast, two haplotypes of FBA10 (Hap1 vs Hap2) showed highly significant differences in heading date phenotypes (Figure 1c in the revised manuscript).” This difference seems statistically supported. So, to be convincing they should show us that these two haplotypes have polymorphisms in FBA10 consistent with the heading date phenotype. In other words, is the predicted amino acid sequence or the expression level of Hap2 of TraesCS3A02G391100/FBA10/qHtL1 consistent with altered activity of the gene and delayed heading? I could not see these data anywhere, and without them I still find the manuscript unconvincing.

Reviewer #2

(Remarks to the Author)

The authors have considered the comments and suggestions that I raised in the first round of review and have modified the manuscript accordingly.

Reviewer #3

(Remarks to the Author)

The authors responded satisfactory to my comments.

Version 2:

Reviewer comments:

Reviewer #1

(Remarks to the Author)

In this second revision, the authors have seriously considered my comments and edited the manuscript accordingly. Including the data on the CRISPR mutants as well as the details of the sequence differences and expression levels of the haplotypes have improved the manuscript and answered my criticisms. Showing that in Hap1 TraesCS3A02G391100/FBA10/qHtL1 is slightly more highly expressed than in Hap2 in supplementary Figure 5 provides a logical basis for the heading date phenotype observed, although in my opinion the difference in mRNA level does look very small. Nevertheless, I realize that the other two reviewers have already accepted the manuscript for publication, and it does now present a logical argument supporting their conclusion that this is the causal gene. Therefore, I support publication of

this version.

Response to Reviewer #1

Reviewer #1 (Remarks to the Author):

This manuscript studies the vernalization response in wheat. It starts with a genetic analysis and identifies a QTL (qHTL1) associated with heading time after vernalization. They identified the causal gene of qHTL1 as FBA10 that encodes the enzyme fructose 1,6-bisphosphate aldolase. Overexpression of FBA10 caused early flowering whereas knock down caused late flowering, and also either increased or decreased aldolase activity. This correlates with increased or reduced levels of VRN1 mRNA, a promoter of flowering after vernalization. They then argue that post-translational modification of FBA10 is altered during vernalization. Phosphorylation of FBA10 is reduced during vernalization whereas O-GluNAcylation is increased. These changes are proposed to increase FBA10 protein stability and lead to changes in histone modification at the VRN1 locus that change its expression.

This manuscript contains a lot of work going all the way from genetic analysis to detailed biochemical and genomic experiments. However, I was not convinced by several of the conclusions and judged that the data shown were insufficient. I realize the difficulties of working in hexaploid wheat, but perhaps a more focused article that made fewer points but examined them more closely would be more convincing. A smaller point is that the writing needs detailed, perhaps professional, editing in many places.

Detailed comments:

1. Resolution of the identification of FBA10. The genetic analysis identifies a large interval including at least 22 genes. The logic for then focusing on FBA10 was suggestive but not totally convincing based on expression level and that the protein is modified during vernalization (lines 122 – 124). Here I wondered what is the difference in sequence between FBA10 alleles, and how does this correlate with gene activity? Also, how many other genes encode FBA10 enzymes in the wheat genome, and how would genetic modifications at this gene affect the total pool of enzyme? Often metabolic enzymes are encoded by genes at multiple loci, making it less clear why modification of a single copy would impact phenotype.

Response:

We appreciate your comment and understand your concerns.

(1) Resolution of the identification of FBA10.

In addition to the expression levels of 22 genes within the significantly associated interval (3A: 639–641 Mb), the association between haplotypes and the phenotype of heading date of potential candidate genes was analyzed. Among the 22 genes, *TraesCS3A02G391200* and *TraesCS3A02G391500* exhibited high expression in spike tissues, while *TraesCS3A02G391100*(FBA10) showed significantly higher expression levels across different spike developmental stages compared to other candidate genes in the interval (please see Figure 1a below, Supplementary Figure 3a in the revised version). It is noteworthy that *TraesCS3A02G391500* encodes aldolase FBA11.

Further haplotype association analysis indicated no statistically significant differences in the trait of heading date between two haplotypes of *TraesCS3A02G391200* (Supplementary Figure 3b in the revised version, Figure 1b below). Similarly, the analysis for *TraesCS3A02G391500/FBA11* haplotypes also demonstrated no significant differences in the heading date trait between the two haplotypes (Supplementary Figure 3c in the revised version, Figure 1c below). In contrast, two haplotypes of *FBA10* (Hap1 vs Hap2) showed highly significant differences in heading date phenotypes (Figure 1c in the revised manuscript). On this basis, the posttranslational modification of *FBA10* in response to vernalization was considered as a clue, indicating a role of *FBA10* in environmental and genetic adaptability.

We further performed Unique Identifier mRNA Sequencing (UID mRNA-seq) to analyze the transcription levels of *TraesCS3A02G391100/FBA10*, *TraesCS3A02G391200* and *TraesCS3A02G391500/FBA11* in response to vernalization in KN199, and found that *TraesCS3A02G391100/FBA10* and *TraesCS3A02G391200* is upregulated following vernalization treatment (V21), whereas *TraesCS3A02G391500/FBA11* is downregulated (Figure 1d below).

Evaluating spatiotemporal expression profiles, haplotype-phenotype associations and expression patterns in response to vernalization together, *FBA10* was identified as the key candidate gene for subsequent functional investigation.

Figure 1. Transcription and haplotype analyses of *TraesCS3A02G391100/FBA10*, *TraesCS3A02G391200* and *TraesCS3A02G391500/FBA11*.

(a) Tissue-specific expression profile of *TraesCS3A02G391100/FBA10* and other candidate genes.

Z: Zadoks decimal code, internationally used to represent developmental stages of cereal crops, with different numbers representing distinct developmental stages. The *TraesCS3A02G391900* was not detected in any wheat tissues, *TraesCS3A02G392500* and *TraesCS3A02G392700* exhibited extremely low or no expression across the tested tissues, thus they were not displayed in the figure.

(b) Heading time analysis between two haplotypes of *TraesCS3A02G391200*^{Hap-1} (n=12 accessions) and *TraesCS3A02G391200*^{Hap-2} (n=58 accessions). The box plots display the interquartile range, comprising the first quartile, median, and third quartile, while the whiskers extend from the minimum to the maximum values. Two-tailed Student's *t*-test for statistical analysis.

(c) Heading time analysis between two haplotypes of *TraesCS3A02G391500*^{Hap-1} (n=13 accessions) and *TraesCS3A02G391200*^{Hap-2} (n=70 accessions). The box plots display the interquartile range, comprising the first quartile, median, and third quartile, while the whiskers extend from the minimum to the maximum values. Two-tailed Student's *t*-test for statistical analysis.

(d) The expression patterns of *TraesCS3A02G391100*, *TraesCS3A02G391200* and *TraesCS3A02G391500* in response to vernalization. KN, wild-type KN199; V0, nonvernalized; V21, vernalized for 21days.

(2) What is the difference in sequence between FBA10 alleles, and how does this correlate with gene activity?

Response:

We conducted a detailed comparison of CDS and promoter sequences among the three alleles, *TraesCS3A02G391100/FBA10*, *TraesCS3B02G423200/FBA13*, and *TraesCS3D02G383700/FBA15* (see Figure 2 and Figure 3 below). Although these three genes share a higher CDS sequence similarity, their promoter regions display distinct variations in motif composition (Table 1 below). Bioinformatics analysis revealed that the promoter region of *FBA10* contains significantly more cis-regulatory elements for binding of MIKC-type MADS and BBR/BPC transcription factor families, compared to *FBA13* and *FBA15*. The MIKC-type MADS-box transcription factor family members (e.g., SOC1, FLC) coordinate signals from the photoperiod pathway and vernalization pathway to regulate transcriptional activation of flowering integrators (e.g., FT, LFY). BBR/BPC transcription factors function in flower development by recruiting Polycomb proteins to mediated H3K27me3 at target genes (Ref#1 and Ref#2 below). These analyses suggest that *FBA10* may function as a central hub in the flowering time regulatory network.

Based on Unique Identifier mRNA Sequencing (UID mRNA-seq) data, these three genes show distinct transcriptional patterns in response to vernalization. Specifically, the transcription level of *FBA10* is the highest both before and after vernalization, with a significant up-regulation observed following 21 days of vernalization. In contrast, vernalization markedly suppresses the transcription of *FBA13*, whereas *FBA15* shows no significant change (see Figure 4 below), indicating that *FBA13* and *FBA15* do not contribute to the requirement of high aldose activity for flowering during vernalization process. Based on the above analysis, the role of *FBA10* in promoting flowering under vernalization conditions is suggested to be independent of the functions of *FBA13* and *FBA15*.

Ref#1: Theune ML, Bloss U, Brand LH, Ladwig F, Wanke D. Phylogenetic Analyses and GAGA-Motif Binding Studies of BBR/BPC Proteins Lead to Clues in GAGA-Motif Recognition and a Regulatory Role in Brassinosteroid Signaling. *Frontiers in plant science* **10**, 466 (2019).

Ref#2: Petrella R, *et al.* BPC transcription factors and a Polycomb Group protein confine the expression of the ovule identity gene SEEDSTICK in *Arabidopsis*. *The Plant journal : for cell and molecular biology* **102**, 582-599 (2020).

TraesCS3A02G391100	ATGTCGGCCTACTGCGGAAAGTACAGGATGAGCTCATCAAGAACGCTGCCTACATGGGCACCCCTGGCAAGGATATCCT	80
TraesCS3B02G423200	ATGTCGGCCTACTGCGGAAAGTACAGGATGAGCTCATCAAGAACGCTGCCTACATGGGCACCCCTGGCAAGGATATCCT	80
TraesCS3D02G383700	ATGTCGGCCTACTGCGGAAAGTACAGGATGAGCTCATCAAGAACGCTGCCTACATGGGCACCCCTGGCAAGGATATCCT	80
Consensus	atgtcggcctactgcggaagtacaaggatgagctcatcaagaacgctgcctacatgggcacccctggcaaggatattcct	
TraesCS3A02G391100	CGCTGCGAGTGAAGTCCACCGGCACCATCGGCAAGCGCTTGCCAGCATCAATGTTGAGAAAGTTGAGGACAAACCGTCCG	160
TraesCS3B02G423200	CGCTGCGAGTGAAGTCCACCGGCACCATCGGCAAGCGCTTGCCAGCATCAATGTTGAGAAAGTTGAGGACAAACCGTCCG	160
TraesCS3D02G383700	CGCTGCGAGTGAAGTCCACCGGCACCATCGGCAAGCGCTTGCCAGCATCAATGTTGAGAAAGTTGAGGACAAACCGTCCG	160
Consensus	gctgc ga gagtccaccggcaccatcggcaagcgctt gccagcatcaa gttgagaa gttgaggacaaccgtcgg g	
TraesCS3A02G391100	CCCTCCGTGAGCTCCTCTTCTGCACCCCTGGCCCTCCAGTACCTCAGCGGGTGATCCTCTTTGAGGAGACCCCTGTAC	240
TraesCS3B02G423200	CCCTCCGTGAGCTCCTCTTCTGCACCCCTGGCCCTCCAGTACCTCAGCGGGTGATCCTCTTTGAGGAGACCCCTGTAC	240
TraesCS3D02G383700	CCCTCCGTGAGCTCCTCTTCTGCACCCCTGGCCCTCCAGTACCTCAGCGGGTGATCCTCTTTGAGGAGACCCCTGTAC	240
Consensus	ccctccgtgagctcctcttctgcacccctgg gccctccagtacctcagcggg gtgatcct tt gaggagaccctgtac	
TraesCS3A02G391100	CAGAGCACCAAGGGTGGCAAGCCCTTCGTCGACATCCTCAAGGGGGCAATGTCTCTCCCGGCATCAAGGTGGACAAGGG	320
TraesCS3B02G423200	CAGAGCACCAAGGGTGGCAAGCCCTTCGTCGACATCCTCAAGGGGGCAATGTCTCTCCCGGCATCAAGGTGGACAAGGG	320
TraesCS3D02G383700	CAGAGCACCAAGGGTGGCAAGCCCTTCGTCGACATCCTCAAGGGGGCAATGTCTCTCCCGGCATCAAGGTGGACAAGGG	320
Consensus	cagagcaccaagggtggcaagcccttcgtcgcacatcctcaagg gggcaa gtctctccc ggcatacaagggtggacaaggg	
TraesCS3A02G391100	TACCTTTCGAGCTTGTGGAACCAACGGTGAAGACCAACCCAGGGCTTTGATGACCTTGGCAAGCGCTGCCAAGTACT	400
TraesCS3B02G423200	TACCTTTCGAGCTTGTGGAACCAACGGTGAAGACCAACCCAGGGCTTTGATGACCTTGGCAAGCGCTGCCAAGTACT	400
TraesCS3D02G383700	TACCTTTCGAGCTTGTGGAACCAACGGTGAAGACCAACCCAGGGCTTTGATGACCTTGGCAAGCGCTGCCAAGTACT	400
Consensus	acc t gagct gctggaaccaacggtagaacac acccagggctttgatgaccttggcaagcgctg gccaaagtact	
TraesCS3A02G391100	ATGAGGCTGGTGCCCGCTTCCCAAGTGGCGTGTCTCCTCAAGATGGGCGCCACAGGCCATCGCAGCTTCCATCGAC	480
TraesCS3B02G423200	ATGAGGCTGGTGCCCGCTTCCCAAGTGGCGTGTCTCCTCAAGATGGGCGCCACAGGCCATCGCAGCTTCCATCGAC	480
TraesCS3D02G383700	ATGAGGCTGGTGCCCGCTTCCCAAGTGGCGTGTCTCCTCAAGATGGGCGCCACAGGCCATCGCAGCTTCCATCGAC	480
Consensus	a gaggctggtgcccgctt gccaaagtggcgtg gtcctcaagat ggc cac gaggcaac gact tccatcgac	
TraesCS3A02G391100	CAGAAAGCTCAGGGTCTGGCTCGATATGCCATCATCTGCCAGGAGAATGGCTGGTCCCATTTGTGAGCCGTGAGATCCT	560
TraesCS3B02G423200	CAGAAAGCTCAGGGTCTGGCTCGATATGCCATCATCTGCCAGGAGAATGGCTGGTCCCATTTGTGAGCCGTGAGATCCT	560
TraesCS3D02G383700	CAGAAAGCTCAGGGTCTGGCTCGATATGCCATCATCTGCCAGGAGAATGGCTGGTCCCATTTGTGAGCCGTGAGATCCT	560
Consensus	cagaa gtcacaggtctggctcg tatgccaatcatctgccaggagaatgg ctggt cccattgt gaggcgtgagatcct	
TraesCS3A02G391100	TGTTGATGGACCTCATGACATTTAGCCGCTGTGCTTGTGTCACCGAGATGTCTCTTGTGCTGCTACAAGGCCTCAACG	640
TraesCS3B02G423200	TGTTGATGGACCTCATGACATTTAGCCGCTGTGCTTGTGTCACCGAGATGTCTCTTGTGCTGCTACAAGGCCTCAACG	640
TraesCS3D02G383700	TGTTGATGGACCTCATGACATTTAGCCGCTGTGCTTGTGTCACCGAGATGTCTCTTGTGCTGCTACAAGGCCTCAACG	640
Consensus	tgttgatggacctcatgacatttagccgctgtgcttgtgtcaccgagatgtctcttgtgctgctacaaggcctcaacg	
TraesCS3A02G391100	ACCAGCATGTCTCCTTGAAGGCAACCTCCTGAAGCCCAACATGTTACCCCTGTTTCCGACTCAAGAAGGTGGCCCT	720
TraesCS3B02G423200	ACCAGCATGTCTCCTTGAAGGCAACCTCCTGAAGCCCAACATGTTACCCCTGTTTCCGACTCAAGAAGGTGGCCCT	720
TraesCS3D02G383700	ACCAGCATGTCTCCTTGAAGGCAACCTCCTGAAGCCCAACATGTTACCCCTGTTTCCGACTCAAGAAGGTGGCCCT	720
Consensus	accagcatgtctcctt gaggg ac ctctcgaagcccaaatggt acccctggttc gac ccaagaaggt gccctt	
TraesCS3A02G391100	GAGGTGATTGCTGAGTACACCGTCCGCACCCCTCCAGAGGACCGTCCCTGCTGCCGTCCCGCCATTGCTTCTCTCTCGG	800
TraesCS3B02G423200	GAGGTGATTGCTGAGTACACCGTCCGCACCCCTCCAGAGGACCGTCCCTGCTGCCGTCCCGCCATTGCTTCTCTCTCGG	800
TraesCS3D02G383700	GAGGTGATTGCTGAGTACACCGTCCGCACCCCTCCAGAGGACCGTCCCTGCTGCCGTCCCGCCATTGCTTCTCTCTCGG	800
Consensus	gaggtgattgctgagtacaccg ccgcacccctccagaggaccgtccctgctgccgtccc gccattgcttctctctc gg	
TraesCS3A02G391100	TGGACAGAGCGAGGAGGAGGCGACCCCTGAACCTGAACGCCATGAACAAGCTCCAGACCAAGAAGCCGTGGAACCTGTCT	880
TraesCS3B02G423200	TGGACAGAGCGAGGAGGAGGCGACCCCTGAACCTGAACGCCATGAACAAGCTCCAGACCAAGAAGCCGTGGAACCTGTCT	880
TraesCS3D02G383700	TGGACAGAGCGAGGAGGAGGCGACCCCTGAACCTGAACGCCATGAACAAGCTCCAGACCAAGAAGCCGTGGAACCTGTCT	880
Consensus	tggacagag gaggaggagcgaccctgaacctgaacgccatgaacaagctccagaccagaagccgtggaacctgtcct	
TraesCS3A02G391100	TCTCCTTCGGGCGTGCCTCCAGCAGAGCACCCCTCAAGGCCCTGGCTGGCAAGCGGAGAACGAGGAGAAGGCCAGGCGG	960
TraesCS3B02G423200	TCTCCTTCGGGCGTGCCTCCAGCAGAGCACCCCTCAAGGCCCTGGCTGGCAAGCGGAGAACGAGGAGAAGGCCAGGCGG	960
TraesCS3D02G383700	TCTCCTTCGGGCGTGCCTCCAGCAGAGCACCCCTCAAGGCCCTGGCTGGCAAGCGGAGAACGAGGAGAAGGCCAGGCGG	960
Consensus	tctccttcggcgctgc ctccagcagagaccctcaaggcctgg c ggcaag cggagaa gaggagaagggccagg g	
TraesCS3A02G391100	GCGTTCCTGGTGAAGTGAAGGCCAAGCTCCGAGGCCACCCCTCGCACCTACAAGGGCGAGCCACCCCTGGTGAAGGCGC	1040
TraesCS3B02G423200	GCGTTCCTGGTGAAGTGAAGGCCAAGCTCCGAGGCCACCCCTCGCACCTACAAGGGCGAGCCACCCCTGGTGAAGGCGC	1040
TraesCS3D02G383700	GCGTTCCTGGTGAAGTGAAGGCCAAGCTCCGAGGCCACCCCTCGCACCTACAAGGGCGAGCCACCCCTGGTGAAGGCGC	1040
Consensus	gcttctctggtaggtgcaaggccaactc gaggccaacctcgg acctacaagggcga gccacct g gaggggcg	
TraesCS3A02G391100	CTCCGAGAGCCTCCAGTCAAGGACTACAAGTACTG	1076
TraesCS3B02G423200	CTCCGAGAGCCTCCAGTCAAGGACTACAAGTACTG	1076
TraesCS3D02G383700	CTCCGAGAGCCTCCAGTCAAGGACTACAAGTACTG	1076
Consensus	ctc gagagcctcca gtcaggactacaagtactg	

Figure 2. CDS sequences alignment of *TraesCS3A02G391100/FBA10*, *TraesCS3B02G423200/FBA13* and *TraesCS3D02G383700/FBA15*.

**Figure 3. Promoter sequences alignment of alleles of *TraesCS3A02G391100/FBA10*,
TraesCS3B02G423200/FBA13 and *TraesCS3D02G383700/FBA15*.**

Table 1 Quantitative analysis of shared motifs in the promoter regions of *TraesCS3A02G391100/FBA10*, *TraesCS3B02G423200/FBA13* and *TraesCS3D02G383700/FBA15*.

Family	TraesCS3A02G391	TraesCS3B02G423	TraesCS3D02G383
	100/FBA10	200/FBA13	700/FBA15
BBR-BPC	16	0	4
MIKC-MADS	21	6	12
C2H2	9	10	5
NAC	9	5	0
Nin-like	1	0	0
Trihelix	2	2	0
Dof	7	4	5
ERF	7	51	23
ARF	5	5	1
B3	2	2	5
CPP	4	0	0
G2-like	5	1	1
MYB	8	3	9
SBP	2	0	0
EIL	2	0	1
HD-ZIP	3	0	7
C3H	1	1	1
bZIP	7	1	8
GATA	1	0	6
MYB_related	1	0	2
E2F/DP	0	1	0
ZF-HD	0	1	1
LBD	0	2	0
HSF	0	3	1
TCP	0	1	3
WRKY	0	0	2
WOX	0	0	1
bHLH	0	0	1
BES1	0	0	4

Figure 4. Transcriptional levels analysis of *TraesCS3A02G391100/FBA10*,

***TraesCS3B02G423200/FBA13* and *TraesCS3D02G383700/FBA15* in response to vernalization.**

Note: Plumules of KN199 with (V21) or without (V0) vernalization were collected for Unique Identifier mRNA Sequencing (UID mRNA-seq). V0, Non-vernalized; V21, 21 days of vernalization. Two-tailed Student's *t*-test for statistical analysis.

(3) How many other genes encode FBA10 enzymes in the wheat genome, and how would genetic modifications at this gene affect the total pool of enzyme?

Response:

As you have noted, the wheat genome contains 21 genes encoding fructose 1,6-bisphosphate aldolase. We obtained Unique Identifier mRNA Sequencing (UID mRNA-seq) data from KN199 and *FBA10*-RNAi plants under V0 conditions, as well as from KN199 under both V0 and V21 conditions. The data were used to investigate the expression changes of these 21 alleles in *FBA10*-RNAi materials and to assess the transcriptional differences in vernalization response.

Among the 21 FBAs identified in wheat, FBA19, FBA20, and FBA21 are classified into Class II, while the remaining 18 are distributed across various clusters within Class I, and have distinct subcellular location patterns (see Ref #1 below). Although these FBAs are involved in distinct biological processes, their collective influence on overall aldolase activity remains largely undetermined. In *Escherichia coli*, the zinc-dependent class II fructose-bisphosphate aldolase (FBA II) accounts for 95 to 100% of the total fructose-bisphosphate aldolase activity (Ref #2 below), indicating the different contribution of aldolases in the total pool of enzyme.

FBA10 shows higher transcriptional activities across different spike developmental stages (Figure 1a above), as well as the whole life cycle (Ref #1 below). In addition, we firstly attempted to generate CRISPR knockout mutants of *FBA10*. However, the knockout mutants displayed lethal phenotypes, indicating a predominant role of FBA10 in regulating growth and development in wheat. Taken together, it is suggested that FBA10 plays a dominant role in the total activity of aldolases pool in wheat.

Ref #1: Lv, G. Y., et al. (2017). Molecular Characterization, Gene Evolution, and Expression Analysis of the Fructose-1, 6-bisphosphate Aldolase (FBA) Gene Family in Wheat (*Triticum aestivum* L.). *Front Plant Sci* **8**: 1030.

Ref #2: Zgiby, S. M., et al. (2000). Exploring substrate binding and discrimination in fructose1, 6-bisphosphate and tagatose 1,6-bisphosphate aldolases. *Eur J Biochem* **267**(6): 1858-1868.

2. The genetic testing of *FBA10* is rather crude (lines 127-129). This is based on very strong overexpression or knock down by RNAi. The knock down could affect other genes as well as the gene of interest and this does not seem to have been tested. It should be possible now even in wheat to make CRISPR alleles of specific genes.

Response:

Thank you for your constructive comments.

Initially, we engineered CRISPR/Cas9-based knockout mutants of *FBA10*. However, the knockout mutants displayed lethal phenotypes and severely impaired developmental processes, ultimately preventing seed production (also described in the revised manuscript). These findings align with previous literature indicating that *FBA* knockdown leads to lethal effects (Ref #3 and Ref #4 below). In *FBA10*-RNAi construct, a 273 bp CDS sequence exhibiting the highest specificity for *FBA10* was selected to minimize off-target effects on homologous genes. The effect of *FBA10* knockdown on other FBAs encoding genes were evaluated in detail (Supplementary Figure 7 and Figure 6 below).

1) Kindly refer to the alignment of *FBA10* RNAi target sequence and 21 homologous *FBA* genes sequences corresponding to the target site (Figure 5 below).

2) Considering that *FBA10* serves as a crucial enzyme in basic metabolic processes, its knockdown could substantially influence the expression of numerous genes across diverse pathways. Consequently, we conducted a detailed analysis to assess how *FBA10* knockdown impacts the transcription levels of other homologous *FBA* genes. Among the 21 aldolase genes, six genes of *FBA11~16* share the greatest sequence homology with *FBA10*. Based on UID mRNA-seq data from KN199 and *FBA10*-RNAi samples of plumules under V0, the expression levels of *FBA11~16* are not significantly affected by *FBA10* knockdown, however, the transcription levels of *FBA4*, *FBA7*, *FBA8*, *FBA9*, and *FBA14* decreased in *FBA10*-RNAi plumules (Figure 6a and b below), in addition to *FBA10*.

3) Further analysis showed that vernalization significantly increases the expression of *FBA10*, *FBA7*, *FBA8* and *FBA9*, decreases *FBA4* expression, and has no significant effect on *FBA14* transcription (Figure 6c below), suggesting that *FBA4* and *FBA14* do not contribute to the enhanced aldolase activity in response to vernalization. Based on the downregulation of *FBA7~9* transcription in *FBA10*-RNAi plumules, we further performed qRT-PCR to detect the expression levels of *FBA7~9* at the stem elongation stage in *FBA10*-RNAi and KN199 with or without vernalization treatment. We found that none of *FBA7*, *FBA8* and *FBA9* were downregulated at the stem elongation stage in *FBA10*-RNAi, moreover, these three genes exhibit similar vernalization response in *FBA10*-RNAi as that in KN199. In contrast, *FBA10* keeps a stable significant downregulation under V0 and V21 conditions (Figure 6c, d and e below).

Taken together, we suggest that the decreased expression of *FBA7*, *FBA8* and *FBA9*

in *FBA10*-RNAi plumules results from development stage-dependent indirect gene expression regulation, but not off-target effects. These results demonstrate that the 273-bp target sequence selected for *FBA10* knockdown achieved the anticipated specificity.

It is reported that the 21 aldolase in wheat are categorized into three major groups, with *FBA10* and *FBA7~9* belonging to distinct clusters, exhibiting different subcellular localization and abiotic stress response patterns. And *FBA7~9* are suggested to play roles in photosynthesis. Moreover, *FBA10* exhibits the highest expression levels during the germination and booting stages among the 21 FBA encoding genes, whereas *FBA7~9* show the lowest expression levels during these two stages (Ref #1 above).

In summary, we propose that *FBA10* is the aldolase that plays a pivotal role in response to vernalization, and the late flowering of *FBA10*-RNAi transgenic plants under vernalization is due to inhibited *FBA10* activity. We further suggest that the vernalization-induced changes in the activity and stability of *FBA10* are specific, leading to phenotypic alterations due to its genetic modification.

We hope that our revisions have appropriately addressed your concerns.

Ref #3: Ritterson Lew, C. and D. R. Tolan (2012). Targeting of several glycolytic enzymes using RNA interference reveals aldolase affects cancer cell proliferation through a non-glycolytic mechanism. *J Biol Chem* 287(51): 42554-42563.

Ref #4: Zhang, C. S., et al. (2017). Fructose-1,6-bisphosphate and aldolase mediate glucose sensing by AMPK. *Nature* 548(7665): 112-116.

Figure 6. Transcription levels analysis of FBAs encoding genes in *qHtL1/FBA10*-RNAi plants.

(a) Volcano plot showing six significantly downregulated FBA encoding genes in nonvernalized (V0) *FBA10*-RNAi plumules compared to KN199. Nonvernalized wild-type KN199 and *FBA10*-RNAi plumules were collected for Unique Identifier mRNA Sequencing (UID mRNA-seq).

(b) Transcriptional levels analysis of *FBA4*, *FBA7*, *FBA8*, *FBA9*, *FBA10*, and *FBA14* indicates a downregulation, as illustrated in (a). Data are means \pm SD, Two-tailed Student's *t*-test was used for statistical analysis.

(c) Expression analysis of *FBA4*, *FBA7*, *FBA8*, *FBA9*, and *FBA14* in response to vernalization in KN199. Plumules of KN199 with (V21) or without (V0) vernalization were collected for Unique Identifier mRNA Sequencing (UID mRNA-seq). Data are means \pm SD, Two-tailed Student's *t*-test was used for statistical analysis.

(d) and (e) qRT-PCR analyses show the relative expression levels of *FBA7*, *FBA8*, *FBA9*, and *FBA10*

mRNA during the stem elongation stage in KN199 and *FBA10*-RNAi plants. KN199 and *FBA10*-RNAi plants were nonvernalized (V0) (**d**) or vernalized for 21 days (V21) (**e**), respectively. Leaves were collected for RNA extraction, and the transcript levels of *FBA*s were normalized to *Actin*, then normalized to KN199 plants. Three biological replicates were performed. Data are means \pm SD, Two-tailed Student's *t*-test was used for statistical analysis.

3. I wondered about the pleiotropy of *FBA10* knock down. Since this is an enzyme involved in primary metabolism, do the knock down plants grow more slowly and does this explain at least partly the delay in heading date?

Response:

Thanks for your comment and we understand your concern. To elucidate the effects of *FBA10* knockdown on wheat growth and development, the morphological phenotypes of KN199 and *FBA10*-RNAi plants at the three-leaf stage was photographed and shown as Figure 7a below. Moreover, we analyzed the number of days required for the stem leaves to reach the 3rd, 5th, 6th, and 12th growth stages in KN199 and *qHtL1*-RNAi lines with 21 days vernalization. Statistical analysis of three biological replicates revealed no significant differences in growth rates of stem leaves between KN199 and *qHtL1*-RNAi plants (Figure 7b below). These data suggest that the significantly delayed flowering phenotype observed in *qHtL1*-RNAi lines under vernalization conditions is not due to growth retardation from *FBA10* knockdown.

Figure 7. Analysis of the influence of *qHtL1/FBA10* knockdown on growth rate of KN199.

(a) Phenotypes of KN199 and *qHtL1*-RNAi plants at the third-leaf stage. Plants were photographed after being transplanted to the greenhouse for an additional 9 days of growth, following 21 days of vernalization. Scale Bar, 10 cm.

(b) Days required for the indicated stem leaves emerging in KN199 and *qHtL1*-RNAi wheat plants. The statistical experiment was performed using three biological replicates, with each replicate comprising a count of 12 to 15 plants. Two-tailed Student's *t*-test was used for statistical analysis.

4. Is the effect on phenotype expected from the effect on gene expression? In Figure 1b-d. I found it surprising that in the OE lines *VRN1* expression was increased 4-5x but the heading time phenotype was very small.

Response:

We performed two more biological replicates to analyze the flowering time phenotype of *qHtL1*-OE3 plants, demonstrating the significant early flowering phenotype of the *qHtL1*-OE3 line in comparison to KN199 (Figure 8a below).

VRN1 serves as a key integrator of vernalization and photoperiod signals, subsequently activating the transcription of *VRN3/FT*. *VRN3/FT* is transported from the leaves to the shoot apical meristem, ultimately facilitating the flowering process in wheat (Ref #5 and Ref #6 below). Therefore, we additionally analyzed the expression levels of *VRN3/FT* in leaves of plants after 30 days of growth in the green house, following a vernalization treatment of 21 days. The transcriptional level of *VRN3/FT* in *qHtL1*-OE3 line was significantly elevated compared to KN199 (Figure 8b below), indicating that *qHtL1* overexpression effectively promotes flowering by activating *VRN1*, which in turn up-regulates *VRN3/FT* transcription.

In addition, we extended vernalization treatment by 30 days for both *qHtL1*-OE3 and KN199 plants, from which we randomly selected fourteen plants and compared their shoot apical morphology and spike developmental stages after a growth of 28 days in the green house. We found that, in comparison to KN199, the transition of the stem apex morphology to the reproductive growth phase was markedly expedited in *qHtL1*-OE3 plants compared to KN199 (Figure 8c below, and supplementary Figure 6e in the revised manuscript).

These findings provide additional evidences that *qHtL1* overexpression promotes vernalization-induced flowering.

We hope these additional experiment results have answered your concern.

Figure 8. *qHtL1* accelerates vernalization-promoted flowering

(a) Analysis of heading time of *qHtL1*-OE3 and KN199 plants. Here we show two additional biological

replicates, in which KN199 and *qHtL1*-OE3 plants are grown under a varying soil condition. The box plots display the interquartile range, comprising the first quartile, median, and third quartile, while the whiskers extend from the minimum to the maximum values, (n=20 plants for each group). Two-tailed Student's *t*-test for statistical analysis.

(b) qRT-PCR analysis shows the relative mRNA abundance of *VRN3/FT* in KN199 and *qHtL1*-OE3 plants. RNA samples were extracted from leaves of KN199 and *qHtL1*-OE3 plants at 30 days after transplanted in the green house, following 21 days vernalization treatment. The data were normalized to *Actin*, then normalized to KN199 plants. Independent biological experiments were repeated three times, data are means \pm SD, Two-tailed Student's *t*-test for statistical analysis.

(c) The shoot apical morphology of KN199 and *qHtL1*-OE3 plants. Plants were vernalized at 4°C for 30 days, then transplanted in the green house. After 28 days of growth, a random sample of 14 plants were selected for observation and photographic documentation. Scale bars, 3 mm.

Ref #5: Xu, S., & Chong, K. (2018). Remembering winter through vernalisation. *Nature Plants*, 4(12), 997-1009..

Ref #6: Niu, De, et al (2024). "A molecular mechanism for embryonic resetting of winter memory and restoration of winter annual growth habit in wheat." *Nature Plants* 10.(1): 37-52.

5. I did not find the conclusion that protein modification affected FBA10 protein stability convincing, because I would have liked to see mRNA analysis so that any transcriptional effect could be assessed. Also, I would have liked to see levels of FBA10 protein relative to modified protein assessed in each of the experiments. In Figure 5g, FBA10 protein levels seem to increase during vernalization, but does the level of FBA10 mRNA or is this increase purely due to enhanced stability? In Figure 3d, the level of phosphorylated protein decreases during vernalization, but in this experiment does the level of protein increase as in 5g or not? I would like to see the level of phosphorylated protein relative to the total amount of FBA10 in the same extracts. Same for other modifications.

Response:

Thanks for your comments and concern.

In the *qHtL1* overexpression construct, *qHtL1*-HA is driven by the constitutive Ubiquitin promoter. The presence of the constitutively expressed *qHtL1* was verified using specific primers including part of the vector sequence. qRT-PCR analysis revealed no significant differences in *qHtL1* transcription levels across various vernalization periods (V0, V7, V14, V21), as illustrated in Figure 9a below, which aligns with expectations. Therefore, the observed variations in *qHtL1*-HA protein levels during vernalization, as detected by the anti-HA antibody, are attributed to differences in protein stability rather than changes in transcription.

The original Figure 3d and Figure 5g employed different normalization methods, leading to potential ambiguity. The original Figure 3d showed the phosphorylation levels of FBA10 across samples with different vernalization durations, with the levels of FBA10-HA normalized for consistency, and the absolute levels of FBA10-HA were not reflected. The original Figure 5g presented the changes in the absolute FBA10-HA levels over different

durations of vernalization treatment, with the levels of total proteins normalized for consistency, showing the gradual increase of constitutively expressed FBA10.

In the revised manuscript, the variations in phosphorylation and O-GlcNAc modification levels of constitutively expressed FBA10, relative to the total FBA10-HA at different periods of vernalization, were presented utilizing the same normalization method. When the total protein levels in the V0, V7, V14, and V21 samples were maintained consistently, the total FBA10-HA exhibited a gradual accumulation with prolonged vernalization duration, and this trend was consistently observed in Figures 3d and 5d in the revised manuscript. Meanwhile, the phosphorylation levels of FBA10 progressively decreased (Figure 9b below, and Figure 3d in the revised manuscript), while the proportion of its O-GlcNAcylation form increased (Figure 9c below, and Figure 5d in the revised manuscript).

The dynamic regulation of O-GlcNAc modification of proteins is controlled by O-GlcNAc transferase (OGT) and O-GlcNAcase (OGA). Upon treatment of cell lysates from *qHtL1*-OE3 plants with PUGNAc, an OGA inhibitor, to elevate the O-GlcNAcylation at total protein levels, the amount of constitutively expressed *qHtL1*/FBA10 remained relatively stable (Figure 5e in the revised manuscript). In contrast, the application of the inhibitor Alloxan, which removes O-GlcNAc modifications, resulted in a reduction of FBA10 levels (Figure 5f in revised manuscript).

Taken together, we suggest that O-GlcNAc modification of FBA10 facilitate to enhance its stability.

Considering the consistent and partly overlapped results, we have replaced the original Figure 3d with Figure 9b below, and replaced the original Figure 5d with Figure 9c below. In the revised manuscript, the partially overlapping original Figure 5d and 5g have been removed.

Figure 9. The phosphorylation and O-GlcNAcylation of qHtL1/FBA10 respond to vernalization.

(a) The relative mRNA abundance of constitutively expressed *qHtL1* in *qHtL1*-OE3 lines. RNA samples were extracted from plumules of wheat plants vernalized for 0, 7, 14 and 21 days, and transcription levels of *qHtL1* were analyzed by qRT-PCR. The data were normalized to *Actin*, then normalized to plants vernalized for V0. Independent biological experiments were conducted three times, data are means \pm SD, Two-tailed Student's *t*-test for statistical analysis.

(b) Vernalization inhibits phosphorylation of qHtL1. Phosphorylation levels of constitutively overexpressed qHtL1 were evaluated during various vernalization periods in *qHtL1*-OE plants. Total proteins were extracted from plumules of wheat plants vernalized at 4°C for 0, 7, 14 and 21 days. Phosphorylation modification of qHtL1 was detected using anti-phosphoSer/Thr antibody, and the overexpressed qHtL1 was detected using anti-HA antibody. Actin was probed as a loading control. The phosphorylation levels were firstly normalized to qHtL1, and then normalized to Actin. The relative protein abundance was quantified using ImageJ software.

(c) Vernalization induces accumulation of O-GlcNAcylated qHtL1 *in vivo*. Chemoenzymatic labelling approach was conducted for analyzing O-GlcNAcylated qHtL1 in wheat. The *qHtL1*-OE plants were vernalized at 4°C for 0, 7, 14, and 21 days, and then for total protein isolation and chemoenzymatic labelling. Constitutively expressed qHtL1 was detected using anti-HA antibody. The qHtL1 levels were normalized to Actin, and then for quantification of the percentage of O-GlcNAcylated qHtL1 relative its total amount. The relative protein abundance was quantified using ImageJ software.

6. A small point is that there are many words and phrases requiring editing. In the first two

lines of the abstract vernalization is spelt wrongly, and “remains blurry” is not really scientific. In line 32 “involved in two natural hybridization” at least lacks a plural. But there are many examples.

Response:

Thanks for your comments. We have carefully polished our manuscript following the suggestions from you and other reviewers, detailed modifications are highlighted in the revised manuscript with blue fonts, and we hope the revised version is now satisfactory.

Response to Reviewer #2

Reviewer #2 (Remarks to the Author):

This study describes a candidate gene identified by GWAS on chromosome of 3A from wheat that potentially underlies variation in heading date (the timing of flowering) amongst a collection of predominantly Chinese wheat varieties – qHtL1.

The authors propose that a fructose 1,6-diphosphate aldolase 10 is the gene underlying the marker trait associations observed in the region of interest (qHtL1). Then, though biochemical characterisations and analyses of heading date and molecular phenotypes of transgenic plants, the authors provide evidence that biochemical interactions between the candidate gene and a kinase (CDPK13) lead to histone modifications of the VRN1 gene that influence heading date.

This is an interesting and worthwhile topic. The authors present a lot of biochemical data to build a mechanistic model to explain at a molecular level the mode of action for qHtL1.

Overall the manuscript is clear. There are a number of areas where more information would strengthen the manuscript and support the overall conclusions.

The most important point to address is the selection of the candidate gene. At the moment the information used to select the candidate gene was (1) expression data, and (2) statistical association. The weakness here is that all genes in that interval will have a statistical association – the GWAS data tells us that, the genes are all closely linked and will share a common statistical association with the heading date trait. So, the statistical evidence creates circular logic in the current version of the manuscript.

Noting also that one other gene in the genetic interval appears to be a known regulator of histone modifications of MADs box genes? (*TraesCS3A02G391200* is related to FLX). It would be good to rule out this gene as a candidate.

Response:

Thanks for your valuable comments and suggestion.

Through genome-wide association study (GWAS) analysis, 22 candidate genes were identified within the chromosomal interval (3A: 639–641 Mb). To investigate their biological functions in spikelet development, we analyzed transcriptome datasets across multiple wheat tissues. The results revealed that *TraesCS3A02G391200* and *TraesCS3A02G391500* exhibited high expression in spike tissues, while *TraesCS3A02G391100/FBA10* showed significantly higher expression levels across different spike developmental stages compared to other candidate genes in the interval (Figure 1a below), suggesting its potential involvement in flowering regulation. It is worthy to note that *TraesCS3A02G391500* encodes aldolase FBA11.

Haplotype association analysis indicated no significant differences in flowering time between the two haplotypes of *TraesCS3A02G391200* and *TraesCS3A02G391500/FBA11*,

respectively (Figure 1b, c below, Supplementary Figure 3). However, two haplotypes of *FBA10* (Hap1 vs Hap2) showed highly significant differences in heading date phenotypes (Figure 1e in the revised manuscript).

We further performed Unique Identifier mRNA Sequencing (UID mRNA-seq) to analyze the transcription levels of *TraesCS3A02G391100/FBA10*, *TraesCS3A02G391200* and *TraesCS3A02G391500/FBA11* in response to vernalization in KN199, and found that *TraesCS3A02G391100/FBA10* and *TraesCS3A02G391200* is upregulated following vernalization treatment (V21), whereas *TraesCS3A02G391500/FBA11* is downregulated (Figure 1d below).

Evaluating spatiotemporal expression profiles, haplotype-phenotype associations and expression patterns in response to vernalization together, *FBA10* was identified as the key candidate gene for subsequent functional investigation.

Figure 1. Transcription and haplotype analysis of *TraesCS3A02G391100/FBA10*, *TraesCS3A02G391200* and *TraesCS3A02G391500/FBA11*.

(a) Tissue-specific expression profile of *TraesCS3A02G391100/FBA10* and other candidate genes.

Z: Zadoks decimal code, internationally used to represent developmental stages of cereal crops, with different numbers representing distinct developmental stages. The *TraesCS3A02G391900* was not detected in any wheat tissues, *TraesCS3A02G392500* and *TraesCS3A02G392700* exhibited extremely

low or no expression across the tested tissues, thus they were not displayed in the figure.

(b) Heading time analysis between two haplotypes of *TraesCS3A02G391200*^{Hap-1} (n=12 accessions) and *TraesCS3A02G391200*^{Hap-2} (n=58 accessions). The box plots display the interquartile range, comprising the first quartile, median, and third quartile, while the whiskers extend from the minimum to the maximum values. Two-tailed Student's *t*-test for statistical analysis.

(c) Heading time analysis between two haplotypes of *TraesCS3A02G391500*^{Hap-1} (n=13 accessions) and *TraesCS3A02G391200*^{Hap-2} (n=70 accessions). The box plots display the interquartile range, comprising the first quartile, median, and third quartile, while the whiskers extend from the minimum to the maximum values. Two-tailed Student's *t*-test for statistical analysis.

(d) The expression patterns of *TraesCS3A02G391100*, *TraesCS3A02G391200* and *TraesCS3A02G391500* in response to vernalization. KN, wild-type KN199; V0, nonvernalized; V21, vernalized for 21days.

A suggestion is to provide a table summarizing sequence expression patterns and natural variation identified in all the candidate genes in that interval.

Response:

We appreciate your valuable suggestion. In accordance with your recommendation, we have incorporated Supplemental Data 2, which presents the expression patterns of the 22 candidate genes, and Supplemental Data 3, which details the natural variations identified within all 22 candidate genes in the interval, into the revised manuscript.

Alternatively, and even more powerful, would be to isolate loss-of-function mutants in the prime candidate gene to provide causal evidence that this gene controls flowering, beyond the transgenic plants used in this study.

Response:

We appreciate your important suggestion and really agree with your comment.

Unfortunately, the *qHtL1* gene mutants that we previously engineered using the CRISPR/Cas9 system displayed a lethal phenotype. These findings align with previous literature indicating that *FBA* knockdown leads to lethal effects (Ref #1 and Ref #2 below). For future functional analyses of other candidate genes, we intend to acquire loss-of-function mutants, following your recommendations, to further enhance the robustness of gene function analysis.

Ref #1: Ritterson Lew, C. and D. R. Tolan (2012). Targeting of several glycolytic enzymes using RNA interference reveals aldolase affects cancer cell proliferation through a non-glycolytic mechanism. *J Biol Chem* 287(51): 42554-42563.

Ref #2: Zhang, C. S., et al. (2017). Fructose-1,6-bisphosphate and aldolase mediate glucose sensing by AMPK. *Nature* 548(7665): 112-116.

1. The abstract could be improved.

Response:

Thank you for your valuable suggestion, with which we are in full agreement. Accordingly,

we have improved the abstract in the revised version of the manuscript.

2. More information is needed about the source of heading-date data used for GWAS. What were the locations and conditions used for these experiments?

Response:

Thanks for your suggestions, phenotypic data of heading time of 91 wheat accessions from different regions over three consecutive years (2014, 2015, and 2016), as well as BLUP value heading time over three years, were combined with resequencing data for GWAS.

3. It was hard to evaluate the importance of qHTL1 relative to overall variation in heading date. A more comprehensive Manhattan plot should be presented to show the complete picture.

Response:

Thank you for your suggestions. The comprehensive Manhattan plot was presented in Supplementary Figure 1 and Figure 2.

4. Similarly, there needs to be more information provided to support the GWAS analysis. What p-value cut-offs were applied? Was correction of population structure applied?

Response:

Thank you for your suggestions. We have revised this section of the methods pertaining to the GWAS in the revised manuscript. The GWAS was performed for heading time traits on 91 wheat accessions by GEMMA (Genome-wide Efficient Mixed Model Association) software according to methods previously described (Ref #1 and Ref #2 below), and using the mixed linear model (MLM) analysis.

$$y = X\alpha + S\beta + K\mu + e$$

where y represents phenotype; α and β are fixed effects representing marker effects and non-marker effects; and μ represents unknown random effects. X , S , and K are the incidence matrices for α , β , and μ , respectively, and e is a vector of random residual effects. The top three PCs were used to build up the S matrix in Plink (version 1.90b6.18). The matrix of simple matching coefficients was used to build up the kinship (K) matrix. The S matrix and kinship (K) matrix were performed for population structure correction.

Significant P value thresholds ($P < 10^{-5}$) were set to control the genome-wide type I error rate. Then we enlarged the candidate region to 1000 kb centered on the GWAS signal peak to identify candidate genes. The GWAS results were visualized using the R software package CMplot (Ref #3 below), which included the creation of QQ plots and Manhattan plots. Additionally, linkage disequilibrium analysis was performed using the PopLDdecay software. (PopLDdecay: a fast and effective tool for linkage disequilibrium decay analysis based on variant call format files.).

Ref #1: Zhou, X. & Stephens, M (2012). Genome-wide efficient mixed-model analysis for association studies. *Nat Genet.* 44,821-824(2012).

Ref #2: Liu, Y, et al. (2023). Genetic basis of geographical differentiation and breeding selection for wheat plant architecture traits. *Genome Biol.* 24(1), 114.

Ref #3: Yin, L. et al. (2021). rMVP: A memory-efficient, visualization-enhanced, and parallel-accelerated

tool for genome-wide association study. *Genom Proteom Bioinf.* 19, 619-628.

5. Line 142. More details need to be provided to show how the connection between qHtL1 and CDPK13 was revealed? The method and the data generated are not explained.

Response:

Thanks for your suggestion. In the revised manuscript, we have enhanced the explanation regarding the identification of TaCDPK13 as the interacting protein of qHtL1. For further details, please refer to lines 178-183 in the revised manuscript.

6. Please check the manuscript for minor errors. For example, in a few instances there are), at ends of sentences, instead of).

I wish the authors the best of luck with this research.

Response:

Thanks for your kind comments. We have carefully polished our manuscript following the suggestions from you and other reviewers, detailed modifications are highlighted in the revised manuscript with blue fonts, and we hope the revised version is now satisfactory.

Response to Reviewer #3

Reviewer #3 (Remarks to the Author):

Yang et al. aim to dissect the mechanisms regulating vernalization for flowering in wheat. Using a genome-wide association study (GWAS), they mapped an important gene associated with flowering time, qHtL1, which encodes fructose 1,6-bisphosphate aldolase (FBA). They showed that overexpression of this gene increased aldolase activity, while RNAi knockdown decreased it, confirming that FBA does indeed have aldolase activity. The overexpression lines showed slightly earlier flowering, while the knockout lines showed delayed flowering, positioning FBA as a key candidate in this region for regulating wheat vernalization using genetic tools.

The authors further investigated the regulation of FBA and demonstrated its modulation by CDPK13, a kinase, and TaOGT, which mediates O-GlcNAcylation. Their biochemical data suggest that vernalization decreases the phosphorylation level of FBA, which in turn increases its activity. Overexpression of CDPK13 delayed flowering, while its mutation resulted in an earlier flowering phenotype. In addition, they showed that FBA undergoes O-GlcNAc modification in vivo, with FBA protein levels increasing upon vernalization. Notably, loss of FBA protein affects the state of histone modification on VRN1 chromatin, thereby regulating VRN1 expression.

Overall, the paper is interesting. However, several figures have conflicting results and some data need further clarification and quality improvement.

1. In Figure 3d, qHtL1-HA protein levels appear to be consistent across the V0, V7, V14, and V21 stages. However, Figure 5g shows conflicting results, indicating higher qHtL1 protein levels at the V21 stage compared to V0. It is unclear which figure accurately reflects the true levels of qHtL1 and this discrepancy needs to be resolved for consistency.

Response:

Thank you for your comments and we understand your concerns.

In the *FBA10-HA* overexpression construct, *FBA10-HA* is driven by the constitutive Ubiquitin promoter. The anti-HA antibody was employed to monitor changes in the levels of constitutively expressed *FBA10-HA* protein across various vernalization durations.

The original Figure 3d and Figure 5g employed different normalization methods, leading to potential ambiguity. The original Figure 3d showed the phosphorylation levels of *FBA10* across samples with different vernalization durations, with the levels of *FBA10-HA* normalized for consistency, and the absolute levels of *FBA10-HA* were not reflected. The original Figure 5g presented the changes in the absolute *FBA10-HA* levels over different durations of vernalization treatment, with the levels of total proteins normalized for consistency, showing the gradual increase of constitutively expressed *FBA10*.

In the revised manuscript, the variations in phosphorylation and O-GlcNAc modification levels of *FBA10*, relative to the total *FBA10-HA* at different periods of vernalization, were presented utilizing the same normalization method. When the total

protein levels in the V0, V7, V14, and V21 samples were maintained consistently, the total FBA10-HA exhibited a gradual accumulation with prolonged vernalization duration. Meanwhile, the phosphorylation levels of FBA10 progressively decreased (Figure 1a below, and Figure 3d in the revised manuscript), while the proportion of its O-GlcNAcylation form increased (Figure 1b below, and Figure 5d in the revised manuscript). With the progressive increase in vernalization time (V0, V7, V14, V21), there was a significant accumulation of the FBA10-HA protein, and this trend was consistently observed in Figures 3d and 5d in the revised manuscript.

In the revised manuscript, we replaced Figure 3d with Figure 1a below, and Figure 5d with Figure 1b below. In the revised manuscript, the partially overlapping original Figure 5d and 5g have been eliminated.

Figure 1 Detection of FBA10/qHtL1 phosphorylation and O-GlcNAcylation levels in wheat with different vernalization treatments.

(a) Vernalization inhibits phosphorylation of qHtL1. Phosphorylation levels of constitutively overexpressed qHtL1 were evaluated during various vernalization periods in *qHtL1*-OE plants. Total proteins were extracted from plumules of wheat plants vernalized at 4°C for 0, 7, 14 and 21 days. Phosphorylation modification of qHtL1 was detected using anti-phosphoSer/Thr antibody, and the overexpressed qHtL1 was detected using anti-HA antibody. Actin was probed as a loading control. The phosphorylation levels were firstly normalized to qHtL1, and then normalized to Actin. The relative protein abundance was quantified using ImageJ software.

(b) Vernalization induces accumulation of O-GlcNAcylated qHtL1 *in vivo*. Chemoenzymatic labelling approach was conducted for analyzing O-GlcNAcylated qHtL1 in wheat. The *qHtL1*-OE plants were vernalized at 4°C for 0, 7, 14, and 21 days, and then for total protein isolation and chemoenzymatic labelling. Constitutively expressed qHtL1 was detected using anti-HA antibody. The qHtL1 levels were normalized to Actin, and then for quantification of the percentage of O-GlcNAcylated qHtL1 relative its total amount. The relative protein abundance was quantified using ImageJ software.

2. The authors argue that CDPK13 regulates qHtL1, and Figure 3d shows decreased phosphorylation levels using an antibody in a Western blot. It is crucial to identify the exact modification sites *in vivo* that show this decrease and to confirm whether these sites correspond to those identified *in vitro* by incubation of TaCDPK13 with qHtL1. If the sites

overlap, this would provide stronger evidence that CDPK13 directly modifies qHtL1 and plays a key role in regulating its phosphorylation, thereby affecting qHtL1 activity. A remaining question, however, is how CDPK13 itself is regulated by vernalization.

Response:

Thanks for your valuable concerns.

(1) To further confirm that TaCDPK13 can catalyze the phosphorylation of qHtL1, we conducted two supplementary experiments:

1) Purified qHtL1-GST and TaCDPK13-GST were incubated for an *in vitro* phosphorylation reaction. The reaction products were subsequently analyzed by mass spectrometry (MS) to identify the phosphorylation sites on qHtL1. The results indicated that TaCDPK13 catalyzed the phosphorylation of qHtL1 at the T33, T35, S330 and S350 sites. Figure 2a and b below show the identification of T35 and S350 by MS analysis (also shown as supplementary Figure 11 in the revised version).

2) We used anti-HA antibody to enrich qHtL1-HA from the cell lysate of nonvernalized *qHtL1*-OE3 plants, and then performed mass spectrometry analysis to detect phosphorylation sites on qHtL1, confirming that T35 and S350 sites undergo phosphorylation *in vivo* (Figure 3a, b below and supplementary Figure 12 in the revised version).

Both methods above identified overlapped phosphorylation sites of T35 and S350 on qHtL1. Thus, our findings substantiate that TaCDPK13 directly catalyze the phosphorylation of qHtL1 *in vitro* and *in vivo*.

Furthermore, phosphorylation of qHtL1 at site S350 was also identified in our previously published article with a study on modification proteomics for vernalization responsiveness in winter wheat (Ref #1 below).

Ref1 #: Xu, S, et al. (2019). The protein modifications of O-GlcNAcylation and phosphorylation mediate vernalization response for flowering in winter wheat. *Plant physiology*, 180(3), 1436-1449

Figure 2. Liquid chromatography-mass spectrometry (LC-MS) identification of phosphorylation at T35 and S350 sites of qHtL1 expressed in *E. coli*.

(a) Identification of phosphorylation at T35 site on qHtL1 using LC-MS. Peptide sequence including T35 site, GILAEDESTGT(35)IGKR.

(b) Identification of phosphorylation at S350 site on qHtL1 using LC-MS. Peptide sequence including S350 site, ANSEATLGTYKGDATLGE GASES(350)LHVK.

Note: Fusion proteins of qHtL1-GST and TaCDPK13-GST were expressed and purified *in vitro*. Subsequently, they underwent an *in vitro* phosphorylation reaction, followed by analysis using liquid chromatography-mass spectrometry (LC-MS).

Figure 3. Liquid chromatography-mass spectrometry (LC-MS) identification of phosphorylation at T35 and S350 sites on qHtL1 in wheat plants.

(a) Identification of phosphorylation at T35 site on qHtL1 using LC-MS. Peptide sequence including T35 site, GILAADESTGT(35)IGK.

(b) Identification of phosphorylation at S350 site on qHtL1 using LC-MS. Peptide sequence including S350 site, ANSEATLGTYKGDATLGEGASE(350)LHVK.

Total proteins were extracted from nonvernalized *qHtL1*-OE3 wheat plants. qHtL1-HA was immunoprecipitated using an anti-HA antibody, and its phosphorylation modification sites were analyzed by LC-MS.

(2) How CDPK13 itself is regulated by vernalization?

We analyzed the transcription levels of *TaCDPK13* during different vernalization periods. The data showed that the transcription levels of *TaCDPK13* progressively increased with the prolonged vernalization duration (Figure 4 below). As the phosphorylation modification level of qHtL1 progressively declines during vernalization, we hypothesize that *TaCDPK13* may also be involved in other signaling pathways. This hypothesis requires further

experimental validation, such as examining the binding affinity between TaCDPK13 and qHtL1 during different vernalization periods.

Figure 4. The relative mRNA abundance of *TaCDPK13* at different vernalization durations in KN199.

RNA samples were extracted from plumules of wheat plants vernalized for 0, 7, 14 and 21 days, and transcription levels of *TaCDPK13* were analyzed by qRT-PCR. The data were normalized to *Actin*, then normalized to plants vernalized for V0. Independent biological experiments were conducted three times, data are means \pm SD, Two-tailed Student's *t*-test for statistical analysis.

3. Figure 4g shows an increase in phosphorylation when purified qHtL1-GST is incubated with TaCDPK13-OE4 cell lysate, indicating a slight increase in qHtL1 phosphorylation. However, the quality of the data is insufficient. Western blot quantification is not convincing and requires multiple replicates for validation. In addition, quantification of phosphorylation by alternative methods would increase the reliability of the results.

Response:

Thank you for your comments. To address your concern, we conducted two additional biological replicates, which demonstrated an increase in qHtL1 phosphorylation level and enhanced data quality, and the representative data was shown as Figure 5 below. We replaced the original Figure 4g with Figure 5 below with a higher quality in the revised manuscript.

To further confirm that TaCDPK13 can catalyze the phosphorylation of qHtL1, we conducted a supplementary experiment to identify phosphorylation sites and quantify phosphorylation levels of qHtL1. Purified qHtL1-GST and TaCDPK13-GST were incubated for an *in vitro* phosphorylation reaction. The reaction products were subsequently analyzed by mass spectrometry to identify the phosphorylation sites on qHtL1. The data indicated that TaCDPK13 catalyzes the phosphorylation of qHtL1 at the S330, S348, S350, Y18, and T35 sites. Furthermore, MS results demonstrated that increasing the amount of TaCDPK13 in the reaction system correspondingly enhances the degree of phosphorylation at these sites (Figure 6 below, and Supplementary Figure 11a).

Western blot together with LC-MS analyses demonstrated that TaCDPK13 is

responsible for qHtL1 phosphorylation.

Figure 5. Overexpressed TaCDPK13 in wheat plants phosphorylates purified qHtL1.

In vivo overexpressed TaCDPK13 increases the phosphorylation level of *in vitro* purified qHtL1. Total proteins were isolated from plumules of TaCDPK13-OE lines and KN199 without vernalization, followed by incubation with purified qHtL1-GST for subsequent phosphorylation analysis. Actin was used as a loading control. The relative protein abundance was quantified using ImageJ software.

Figure 6. Detection of variation of qHtL1 phosphorylation levels by LC-MS.

The qHtL1-GST and TaCDPK13-GST fusion proteins were obtained using a prokaryotic expression system, and then purified for *in vitro* phosphorylation assay. The qHtL1-GST fusion protein was mixed with different amounts of TaCDPK13-GST, and phosphorylation levels of qHtL1 in each reaction were evaluated by mass spectrometry. In f1, no TaCDPK13-GST was added in the reaction system. From f2 to f4, the amounts of TaCDPK13-GST were gradually increased in the reaction system. By maintaining consistent quantities of qHtL1-GST in each reaction, the amounts of four peptides containing identified phosphorylation sites were observed to increase in response to elevated levels of TaCDPK13-GST within the reaction systems. Colored squares indicate varying degrees of phosphorylation, with dark blue denoting low phosphorylation levels and deep red signifying high phosphorylation levels.

4. Figure 5 illustrates the regulation of qHtL1 by TaOGT1; however, it is unclear how many replicates were performed to ensure the reproducibility of the data. Furthermore, there seems to be an inconsistency in the interpretation of the data between Figure 5g and Figure 5d. If Figure 5g shows a multi-fold increase in qHtL1-HA protein levels following vernalization treatment, how does this correlate with the consistent input levels shown in Figure 5d? This discrepancy raises questions about the experimental design and normalization methods used. In addition, the number of replicates performed and the

reproducibility of these results are not clearly stated, which needs to be addressed to increase confidence in the results.

Response:

Thanks for your constructive comments. As you noted, different normalization methods were used in the original Figure 5g and 5d. In the original Figure 5d, the amounts of qHtL1-HA under V0 and V21 conditions were normalized to be consistent to compare the levels of O-GlcNAc modified qHtL1-HA. In the original Figure 5g, the total protein levels corresponding to V0, V7, V14 and V21 were maintained consistent and normalized to Actin, showing a gradual increase of qHtL1-HA following extended vernalization durations.

We performed three biological replicates, which demonstrated that the levels of qHtL1-HA progressively increased with the duration of vernalization treatment. In addition, our results from two biological replicates demonstrated that the O-GlcNAc modification degrees of qHtL1-HA gradually increased following extended vernalization durations, which accompanied the elevation of total qHtL1-HA levels. The representative result is presented in Figure 7 below and in Figure 5d in the revised manuscript.

Considering consistent and partly overlapped results, the original Figure 5d and 5g were removed in the revised manuscript.

Figure 7. Vernalization induces accumulation of O-GlcNAcylated qHtL1 *in vivo*.

Chemoenzymatic labelling approach was conducted for analyzing O-GlcNAcylated qHtL1 in wheat. The *qHtL1*-OE plants were vernalized at 4°C for 0, 7, 14, and 21 days, and then for total protein isolation and chemoenzymatic labelling. Constitutively expressed qHtL1 was detected using anti-HA antibody. The qHtL1 levels were normalized to Actin, and then for quantification of the percentage of O-GlcNAcylated qHtL1 relative its total amount. The relative protein abundance was quantified using ImageJ software.

5. If Figure 5g shows that the qHtL1 level increases several fold between v0 and v21, it is puzzling why the activity of the aldolase level increases only very slightly (900 to 1200?). To prove the model, is it possible to treat the wheat with the product (G3P) of qHtL1 and show that it would affect the histone modification states on VRN1 and affect the flowering time?

Response:

Thanks for your comments and valuable suggestions.

(1) In enzymatic reactions, the augmentation of protein concentrations generally does not demonstrate a linear correlation with the enhancement of enzyme activity. Despite a significant elevation in the protein level of qHtL1-HA relative to the V0 condition following 21 days of vernalization, the enzyme activity did not exhibit a proportional increase. We postulate that this discrepancy is attributable to an insufficient rise in substrate levels.

According to Figure 3f (original and revised versions), 21 days vernalization treatment resulted in a ~30% increase in aldolase activity in wheat *qHtL1*-OE plants compared to wild-type KN199, lead to early flowering phenotype. The degree of variation in qHtL1 enzyme activity resulting in phenotypic modifications aligns with findings reported in a literature, in which, a 33% reduction in aldolase activity significantly affects the intracellular growth of *Salmonella* (Ref # below).

(2) We agree with your suggestion that, exogenous application of G3P, the product of qHtL1, should have an impact on histone modification states at *VRN1* locus and flowering time. It is reasonable to speculate that the substrate of qHtL1, FBP (fructose 1,6-bisphosphate), also possibly exert similar influence with G3P. We treated KN199 plants with 25mM and 50mM FBP during vernalization period of V21, and found that FBP treated plants showed an early flowering time phenotype comparing with untreated control plants (Figure 8a, b below).

In addition, we further evaluate the role of FBP application on histone modification states at *VRN1* in KN199 wheat plants vernalized for 21 days (V21). The KN199 material, three days post-germination, was subjected to a 21-day vernalization treatment. FBP was applied at 24 hours prior to sampling. H3K27ac and H3K14ac modification levels at region IV of the *VRN-A1* locus were evaluated by CHIP-qPCR analysis, demonstrating that FBP addition resulted in a significant increase in these two regions (Figure 8c, d below). Moreover, quantitative PCR (qPCR) assays demonstrated a significant upregulation of the transcriptional level of *VRN1* in plants treated with FBP in comparison to the untreated control (Figure 8e), indicating the role of H3K27ac and H3K14ac in activating *VRN1*.

Together these results support our proposed model in Figure 6f.

Ref #: Fitzsimmons, L., et al. (2018). "Zinc-dependent substrate-level phosphorylation powers *Salmonella* growth under nitrosative stress of the innate host response." *PLoS Pathog* 14(10): e1007388.

Figure 8. FBP treatment enhances H3K14ac and H3K27ac modifications levels at *VRN1*.

(a) Exogenous addition of FBP accelerates flowering in KN199. KN199 was treated with 25 mM and 50 mM FBP during vernalization process, respectively. Wild-type KN199 plants and FBP-treated KN199 plants were transplanted in a green house, following a 21-day vernalization treatment. Scale bar, 10 cm.

(b) Heading time statistical analyses of (a). The box plots display the interquartile range, comprising the first quartile, median, and third quartile, while the whiskers extend from the minimum to the maximum values, ($n=15$ plants for each group). Two-tailed Student's *t*-test for statistical analysis.

(c) and **(d)** ChIP-qPCR assays of H3K14ac and H3K27ac levels in the region IV at *VRN1*. Schematic diagram of *VRN1*, showing the region detected in ChIP-qPCR analysis. Two biological replicates were performed, with each biological replicate comprising three technical replicates. Two-tailed Student's *t*-test for statistical analysis.

(e) qRT-PCR assay shows that exogenous addition of FBP activates *VRN1* transcription. The KN199 plants, 3 days after germination, were subjected to a 21-day vernalization treatment. FBP was applied at 24 hours prior to sampling. Data are means \pm SD of three biological replicates. Two-tailed Student's *t*-test was used for statistical analysis.

6. Additional Points:

All figures from Figure 1 through Figure 6 use traditional bar graphs. To improve visual clarity and allow for a better understanding of the actual data distribution, box plots or beeswarm plots should be used.

Response:

Thank you for your valuable suggestion. In the revised manuscript, all bar graphs have been converted to box plots, with the exception of the qPCR results, UID mRNA-seq data

and results of aldolase activity assay, which remain unchanged due to only three data points.

Response to Reviewer #1

Reviewer #1 (Remarks to the Author):

However, I remain unconvinced that they have proven that TraesCS3A02G391100/FBA10/qHtL1 is the causal gene for the heading date phenotype. For me, key issues are to do with the functional differences between the haplotypes and the genetic confirmation that this is the causal gene. The genetic confirmation is complicated by the result they mention in the response document that a CRISPR-induced allele is lethal – so they cannot test its effect on heading date. I think that they should include this CRISPR allele and the lethality phenotype in the manuscript, so that the reader knows that this experiment was done. In the response document, they go on to say “In contrast, two haplotypes of FBA10 (Hap1 vs Hap2) showed highly significant differences in heading date phenotypes (Figure 1c in the revised manuscript).” This difference seems statistically supported. So, to be convincing they should show us that these two haplotypes have polymorphisms in FBA10 consistent with the heading date phenotype. In other words, is the predicted amino acid sequence or the expression level of Hap2 of TraesCS3A02G391100/FBA10/qHtL1 consistent with altered activity of the gene and delayed heading? I could not see these data anywhere, and without them I still find the manuscript unconvincing.

Response:

Thank you for your comments and we greatly appreciate your hard work and constructive suggestions, which greatly helped us to improve the manuscript.

(1) We had previously attempted to simultaneously disrupt all A-, B- and D-homoeologs of *FBA10/qHtL1* in the KN199 background utilizing the CRISPR/Cas9 genome editing approach, and three single-guide RNAs (sgRNAs) were designed to target different sites in exon 2, respectively, which was shown as Supplementary Figure 7 in the revised manuscript, as well as Response Figure1a and Response Figure2 below. PCR analysis utilizing primers specifically designed to amplify the *Cas9* gene successfully identified four transgenic plants among the 32 T0 generation plants (please see Response Figure1b below). Two independent mutant lines, *cr-1* and *cr-2* were identified by further sequencing analyses. Sequencing results revealed that the *cr-1* mutant exhibits a base substitution at 178 bp downstream of the ATG start codon within the 3A genome, accompanied by a fragment deletion spanning from 181 bp to 248 bp. In the 3B genome, a fragment deletion is detected between 180 bp and 246 bp downstream of the ATG. In the 3D genome, a fragment deletion occurs between 177 bp and 330 bp downstream of the ATG. The *cr-2* mutant demonstrated three distinct fragment deletions: one located between 181 bp and 248 bp downstream of the ATG start codon in the 3A genome, another spanning from 181 bp to 246 bp downstream of the ATG in the 3B genome, and a third ranging from 177 bp to 330 bp downstream of the ATG in the 3D genome. These data are presented in Supplementary Figure 7 of the revised manuscript, as well as Response Figure 1a and Response Figure2 below.

We obtained only the two mutant lines previously mentioned, despite three transformation attempts had been conducted to generate *FBA10/qHtL1* knockout mutants by CRISPR-Cas9

system. The subsequent two attempts even failed to yield any positive transgenic plants. These two mutants exhibited markedly impaired growth and developmental phenotypes, ultimately resulting in lethal phenotype within two months. Unfortunately, we did not capture photographic documentation of the phenotypes of these mutants at that time. We suggest that the complete loss of *FBA10/qHtL1* function results in a lethal phenotype. Consequently, we proceeded to construct RNAi materials targeting the *FBA10/qHtL1* gene. The related information has been incorporated into the revised manuscript.

Response Figure 1 Generation and molecular identification of *FBA10* knockout mutants

(a) Three single guide RNA (sgRNA) sequences are designed to target regions within exon 2 of the three *TraesCS3A02G391100/FBA10/qHtL1* homoeologs. In the diagram, exons are depicted as gray rectangles, and the positions of the target sites are indicated by red bold lines. Sequencing results highlight a base substitution in red and base deletions as red dashes, with numbers on the right indicating gap lengths.

(b) PCR analysis to identify the presence of *Cas9* in T0 plants produced via CRISPR-Cas9 genome editing. *Cas9*-specific primers were designed to amplify a 338 bp fragment. A total of 32 T0 seedlings were obtained and screened, among which lines 1, 2, 16, and 30 were identified as positive transgenic plants. These lines were subsequently subjected to sequencing analysis to confirm the mutation patterns.

Response Figure 2 Identification of *cr-1* and *cr-2* via sequencing

Notes: In the diagram, exons are depicted as gray rectangles, and the positions of the target sequences are indicated by red bold lines. The target site sequences for the three sgRNAs and the mutation patterns in *cr-1* and *cr-2* mutants are shown. Sequencing results highlight a base substitution in red and base deletions as red dashes, with numbers on the right indicating gap lengths.

(2) We performed a comprehensive analysis of Hap-1 and Hap-2, revealing that most SNPs are situated within the promoter region (3A: 639082776-639084776) of the *TraesCS3A02G391100/FBA10/qHtL1* gene. Utilizing the Binding Site Prediction tool available on the PlantRegMap platform (<https://plantregmap.gao-lab.org/index.php>), we identified that the SNP S3A_639082983 is located within the binding site (CACACACACTCTATCTTTA) of the BBR-BPC transcription factor (Supplementary Data 4 in the revised manuscript). This binding site has been previously validated to regulate gene expression in barley, as documented in the literature (Ref#1 below). Consequently, we propose that, despite the absence of amino acid sequence alterations between Hap-1 and Hap-2, the SNP variation within the promoter region may influence the recognition of the BBR-BPC transcription factor, thereby modulating the expression level of *FBA10/qHtL1*. The information has been included in the revised manuscript.

It has been reported that *FBA10/qHtL1* exhibits high expression levels at different development stages compared with its homologous family members, especially at germination

and inflorescence emergence stages (Ref#2 below). We performed a new additional qRT-PCR analysis to examine the differential expression of *FBA10/qHtL1* in wheat plumules. This analysis encompassed 12 accessions with the Hap-1 haplotype and 23 randomly selected accessions with the Hap-2 haplotype. The experiment was conducted with three independent biological replicates. The qRT-PCR results indicated that the expression levels of *FBA10/qHtL1* are generally elevated in the Hap-1 group relative to the Hap-2 group, aligning with the phenotypic differences in heading time observed between the two haplotypes. The qRT-PCR data were shown as Supplementary Figure 5 in the revised manuscript and Response Figure 3 below. This result, in conjunction with other analyses, suggests that the predicted *TraesCS3A02G391100/FBA10/qHtL1* is the candidate gene for the genomic region identified through GWAS analysis. The related information has been included in the revised version of the manuscript.

Ref#1: Santi L., et al. (2003). The GA octodinucleotide repeat binding factor BBR participates in the transcriptional regulation of the homeobox gene *Bkn3*. *Plant Journal* 34(6): 813-826.

Ref#2: Ref #1: Lv, G. Y., et al. (2017). Molecular characterization, gene evolution, and expression analysis of the fructose-1, 6-bisphosphate aldolase (FBA) gene family in Wheat (*Triticum aestivum* L.). *Front Plant Sci* 8: 1030.

Response Figure 3 Expression analysis of *FBA10/qHtL1* in Hap-1 and Hap-2 groups.

Notes: The transcript levels of *FBA10/qHtL1* in plumules of 12 Hap-1 and 23 randomly selected Hap-2 accessions were analyzed by qRT-PCR. Three independent biological replicates were conducted, and the data from all replicates are presented in the figure. Data are means \pm s.d., two-tailed Student's *t*-test was used to determine the *P* value.

Response to Reviewer #1

Reviewer #1 (Remarks to the Author):

In this second revision, the authors have seriously considered my comments and edited the manuscript accordingly. Including the data on the CRISPR mutants as well as the details of the sequence differences and expression levels of the haplotypes have improved the manuscript and answered my criticisms. Showing that in Hap1 TraesCS3A02G391100/FBA10/qHtL1 is slightly more highly expressed than in Hap2 in supplementary Figure 5 provides a logical basis for the heading date phenotype observed, although in my opinion the difference in mRNA level does look very small. Nevertheless, I realize that the other two reviewers have already accepted the manuscript for publication, and it does now present a logical argument supporting their conclusion that this is the causal gene. Therefore, I support publication of this version.

Response:

We would like to express our sincere gratitude for your constructive feedback and for endorsing the acceptance of our manuscript. Your insights have been instrumental in helping us improve our work to a deeper level.